# Integrated cohort of esophageal squamous cell cancer reveals genomic features underlying clinical characteristics

Minghao Li[1,5], Zicheng Zhang[2,3,5], Qianrong Wang[4], Yan Yi[3] & Baosheng Li [1,3] ✉

Esophageal squamous cell cancer (ESCC) is the major pathologic type of esophageal cancer in Asian population. To systematically evaluate the mutational features underlying clinical characteristics, we establish the integrated dataset of ESCC-META that consists of 1930 ESCC genomes from 33 datasets. The data process pipelines lead to well homogeneity of this integrated cohort for further analysis. We identified 11 mutational signatures in ESCC, some of which are related to clinical features, and firstly detect the significant mutated hotspots in *TGFBR2* and *IRF2BPL*. We screen the survival related mutational features and found some genes had different prognostic impacts between early and late stage, such as *PIK3CA* and *NFE2L2*. Based on the results, an applicable approach of mutational score is proposed and validated to predict prognosis in ESCC. As an open-sourced, quality-controlled and updating mutational landscape, the ESCC-META dataset could facilitate further genomic and translational study in this field.

Esophageal squamous cell cancer (ESCC) arises from the epithelial cells of the esophagus and presented typical features of squamous cell carcinoma, which is the major pathologic type of esophageal cancer in Asian population[1]. Since 2012, there had been dozens of investigations published using the whole-genome sequence (WGS) or whole-exome sequence (WES) strategy to explore the genetics of ESCC. These studies depicted the general mutational landscape of ESCC, including the significantly mutated genes such as *TP53*, *CDKN2A*, *EP300*, *PIK3CA*, and *NOTCH1*, the commonly influenced pathways such as PI3K-AKT axis, cell cycle, and histone modification, and the commonly identified age-related and APOBEC enzymes-related mutational signatures[2–20].

However, in the analysis of clinical variables-related genomic features, which is essential for translational research, many previously reported results were contradictory. The high genomic heterogeneity of ESCC and the sample size in a single dataset limited the statistical power in detailed comparisons. The integration of multi-source genomic and clinical data that could provide a more detailed mutational atlas, especially for events with low frequency, might be a solution to the problem, whereas the data-source-associated confounding factors must be well identified and controlled.

Here, we show a quality-controlled integrated ESCC genomic dataset of ESCC-META cohort, and based on it, we systematically evaluate the genomic features underlying clinical characteristics.

## Results

### Overview of ESCC-META cohort

To build the integrated tumor-type-specific genomic cohort, we established a set of pipelines for data selection and process (see Methods for details). Currently, we had integrated 1930 ESCC genomes from 33 datasets, including our own sequence cohort of ECRT (*n* = 42, Fig. 1a). Among them, 413 patients from 15 datasets (including our own sequence data) were reanalyzed from raw reads data, and the rest somatic mutational records (1517 patients from 18 datasets) were prepared from the published mutational list (Supplementary Data 1,2 and Supplementary Table 1). With enormous efforts in data processing and verification, we minimized the potential influence of the

[1]Cheeloo College of Medicine, Shandong University, 250012 Jinan, Shandong, China. [2]Department of Radiation Oncology, Shenzhen Traditional Chinese Medicine Hospital, The Fourth Clinical Medical College of Guangzhou University of Chinese Medicine, Shenzhen, China. [3]Shandong Cancer Hospital and Institute, Shandong First Medical University and Shandong Academy of Medical Sciences, Jinan, China. [4]The Third Affiliated Hospital of Shandong First Medical University, Jinan, Shandong Province, China. [5]These authors contributed equally: Minghao Li, Zicheng Zhang. ✉e-mail: bsli@sdfmu.edu.cn

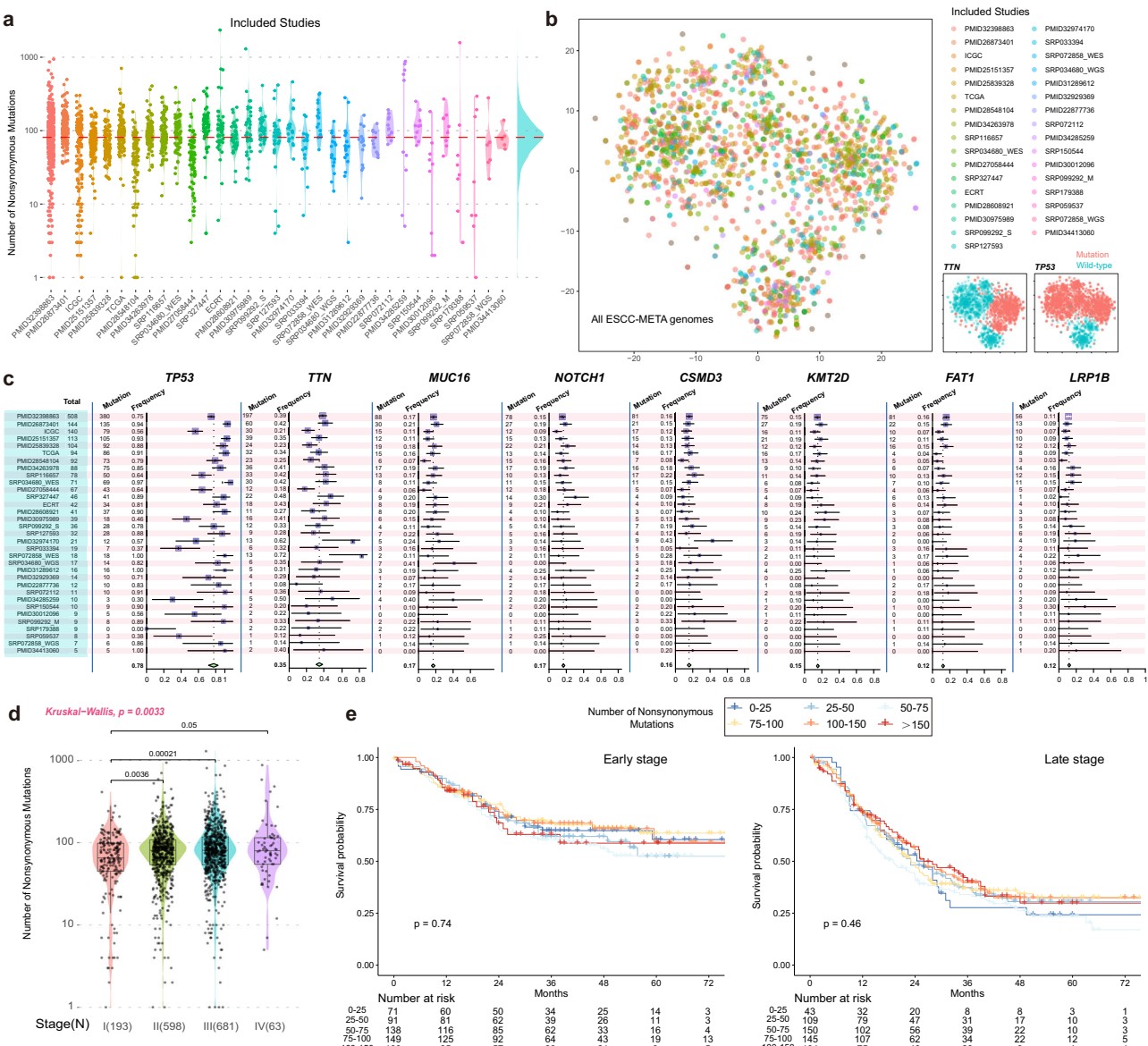

**Fig. 1 | Overview of the ESCC-MATA cohort. a** All of the included studies and the number of the nonsilent mutations in the ESCC-META cohort. The datasets were ranked by their sample size from left to right. The red horizontal line indicated the median number in overall genomes. **b** The scatter plot of all genomes by t-SNE analysis. The dots were colored by datasets (left) or mutational status of *TTN* and *TP53* (right). The t-SNE analysis was performed by the mutation matrix of all integrated genomes of the top 1000 genes. **c** forest plot of the mutational frequency for most common genes in ESCC among all included datasets. The total number of patients in each dataset was labeled in the leftmost panel (blue region). The gene-specific mutated numbers and frequencies in each dataset were presented in the left panel of the gene-specific region. The corresponding forest plots were in the right part. The error band for each line in the forest plot represents the 95% confidence interval of mutational frequency. **d** Comparison of the mutational load

between different tumor stages. All the patients with available stage information were involved in this comparison. The Krustal−Wallis test was used to estimate the significance among the four groups, and the Wilcoxon test was used to estimate the difference in two groups comparison. In the boxplot, the lower extreme line, lower end of box, inner line of box, upper end of box and upper extreme line represent the value of (Q1 − 1.5×IQR), Q1, Q2, Q3 and (Q3 + 1.5×IQR), respectively. Q1−25th quartile; Q2−50th quartile or the median value; Q3−75th quartile. The interquartile range (IQR) is distance between Q1 and Q3 (Q3 − Q1). **e** Survival comparison between different mutational loads in both early-stage (stage I or II) and late-stage (stage III or IV) patients. All the patients with available stage information were involved in this comparison. The log-rank method was used to estimate the significance. Source data are provided as a Source Data file.

heterogeneities in data sources, sequence strategy, and analysis methods among datasets (see Methods for details).

In the tSNE dimensionality reduction analysis, the distributions of clusters were mainly shaped by frequently mutated genes, while no obvious batch effects among datasets could be observed (Fig. 1b). The most frequently mutated genes would not necessarily suggest their important contributions in ESCC tumorigenesis, because many of them might owe to their great coding lengths, such as *TTN* and *MUC16* (Supplementary Fig. 1b). However, their mutational among cohorts

could be used to assess the homogeneity among datasets. We examined the mutational frequencies of the most frequently mutated genes in ESCC, including *TP53* (78%), *TTN* (35%), *MUC16* (16%), *NOTCH1* (16%), *CSMD3* (15%), *KMT2D* (11%), *FAT1* (10%), and *LRP1B* (10%). These genes were generally ranked among the top mutated genes in single datasets, and their cumulative mutational frequencies in the overall dataset were very close to the pooled mutational frequencies calculated by inverse variance weighted estimation (Fig. 1c, see Methods for details), which suggested well homogeneity in commonly mutated genes. We

further detected potential explanatory variables to a load of non-synonymous mutations by multivariate regression analysis. The results indicated that, apart from one dataset that might be influenced by stochastic sampling error in a small sample size (PMID30012096, $n = 9$), the sources of genomes did not significantly influence mutational load (Supplementary Fig. 1a). The median number of non-synonymous mutations in the ESCC-META was 81 (52 of 25th percentiles and 117 of 75th percentiles). Both the multivariate regression analysis and the comparative test indicated stage I patients had significantly lower mutational loads than higher stage (Supplementary Fig. 1a and Fig. 1b). However, in either early or late stage, patients with varied mutational loads did not suggest different prognosis (Fig. 1e).

The WGS and WES sequence types did not significantly influence the detected mutational load, but the heterogeneity among capture platforms of included WES studies deserved further evaluation. The WES sequence platforms were designed to capture total coding regions but would significantly change with the updated genomic annotations[21,22], which might bring bias in mutations located in varied capture ranges. We used 642 WGS sequenced genomes as the test set to estimate the percentage of uncaptured nonsilent mutations in different capture platforms. No more than 1% of nonsynonymous SNVs in the test set would be dropped in varied WES capture platforms, suggesting few biases brought by the heterogeneous capture platforms. Most of the influenced nonsilent SNVs were rare mutations or mutations annotated in splicing sites, while the total coding regions of several genes with potential research values were not fully covered in some platforms, including *MUC4*, *OR2L8*, and *AP3S1*. We listed these genes (Supplementary Fig. 2b and Supplementary Data 3) and reminded readers that their mutational frequencies might be underestimated.

Due to the heterogeneity in the sequence methods of our included studies, we did not provide the estimation of tumor mutational burden (TMB), which was greatly influenced by total capture length (as the denominator in its calculation) and would be misleading in direct comparisons between different platforms[23].

Based on the above analyses, we thought this integrated genomic cohort could be jointly used for further analyses. This integrated dataset was named as ESCC-META cohort, which was aimed to provide a systematic, open-source, and updating genomic resource for researchers in this field.

## Integrated mutational signature analysis

Although some previous ESCC mutational signature analyses were based on WES data, the WGS could provide much more mutational records to estimate mutagenesis. We, therefore, used mutational results from WGS ($n = 1084$) as a discovery set to perform de novo mutational signature analysis, which included 532 genomes in ESCC-META dataset (from PMID32398863, SRP034680_WGS, and SRP072858_WGS) and a newly published SBS96 matrix of 552 ESCC patients[24] (Supplementary Data 4). The median number of total base substitutions was 10,658 in the discovery set without data-source-related divergence (Fig. 2a), and the t-SNE analysis of the matrix of the 96 mutational types also indicated no obvious batch effect among the four studies (Fig. 2b). Notably, the batch effects were obvious in terms of extracted 83 features of small insertions and deletions (ID83, Supplementary Fig. 3a, b). We do not have effective approaches to suppress these batch effects and thus exclude them from the current analysis.

We applied non-negative matrix factorization algorithm (NMF) to identify prominent mutational signatures in the discovery set. The optimal number of separations (K = 11) was selected both considering cophenetic correlations and residual sum of squares (Fig. 2c, see Methods)[25], and the 11 extracted signatures were named from sig1 to sig11 (Supplementary Data 5). We then deconvolved the contributions of the 11 signatures in both the WGS cohort (Fig. 2d) and total

ESCC-META patients (Fig. 2f). We could see that the top 5 signatures (sig1, sig2, sig4, sig6, and sig8) dominated 91.8% of all patients (Fig. 2e). In the WGS cohort, the contributions were significantly related to the source of ESCC genomes in sig5 and sig6, but not in sig1 or sig2 (Fig. 2d). The sig5 were similar to SBS17b (cosine similarity = 0.92, Fig. 2e, Supplementary Data 6) and the sig6 was matched to SBS18 (similarity = 0.98), both of whom were related to damage by reactive oxygen species.

This sig1 featured by evaluated T > C mutations (Fig. 2f) and was similar to SBS16 (similarity = 0.88) or SBS5 (similarity = 0.82), whose aetiologies were unclear. Given that the contributions of sig1 were significantly higher in patients with a smoking or drinking history (Fig. 2i), we speculated this type of mutagenesis might be related to alcohol or tobacco exposure. The sig1-dominated patients (cluster1) presented a significantly worse prognosis compared with other signatures in single variable comparison (Fig. 2j) or multivariable-adjusted Cox regression (hazard ratio = 1.37, *p*-value = 0.016, Supplementary Fig. 3e).

The sig2 was a major mutational contributor in 44.7% ESCC genomes and well matched to SBS1 (similarity = 0.92), which was caused by spontaneous deamination of 5-methylcytosine. In consistent with the age-related accumulations of the mutational process, we observed a significant association between the diagnostic age of ESCC and the contribution of sig2 (Fig. 2h).

The sig7 was similar to SBS2 (similarity = 0.99), and the sig8 was similar to SBS13 (similarity = 0.92), which could be attributed to the activity of APOBEC enzymes (apolipoprotein B mRNA editing enzyme, catalytic polypeptide-like)[26]. The APOBEC-related signatures explained major mutagenesis in 16.8% ESCC genomes, and their contributions were significantly positively correlated with mutational load (Supplementary Fig. 3c, d).

The sig4 presented a nearly even distribution of the 96 types of base substitutions and was similar to SBS3 (similarity = 0.90), which SBS3 was thought to be associated with failure of DNA double-strand break repair. However, in the ESCC-META cohort, the percentage of sig4 presented a negative correlation with mutational load (Supplementary Fig. 3d) and was also unrelated to somatic BRCA1/2 mutations (Supplementary Fig. 3e). The sig3 was to SBS15 (similarity = 0.96) and the sig11 (dominated in 1.2% patients) was similar to SBS44 (similarity = 0.88) and SBS20 (similarity = 0.79), all of whom were associated with DNA mismatch repair.

There were 1.2% patients ($n = 24$) who presented a prevalent mutational pattern of sig9 or SBS22 (similarity = 0.98), which was associated with aristolochic acid exposure and thus suggested the specific carcinogenesis in this subgroup patients[27,28].

## Functional summary of mutational profiles

We summarized the mutated genes by their related oncogenic pathways (Fig. 3a) and found that 38.1% ESCC patients had at least one mutation in Hippo pathway (including *FAT1*, *FAT2*, and *FAT3*), 38.6% in histone modification, 33.8% in NOCTH pathway (*KMT2D*, *KMT2C*, *EP300*, and *CREBBP*), 19.8% in RTK-RAS pathway (*ERBB4* and *ROS1*), 17.6% in cell cycle pathway (*CDKN2A*, *RB1*), 15.3% in PI3K pathway (*PIK3CA*), and 12.6% in Nrf2 pathway (*NFE2L2*, *KEAP1*). While the majority of total nonsilent mutations belonged to missense mutations (84.8%), some genes presented a high chance of truncating mutations (nonsense mutations or frameshift INDELs), including cell-cycle-related genes of *CDKN2A* (located in 9p21, 73.0% mutations were truncating) and *RB1* (in 13q14, 83.9% truncating), Notch pathway-related genes of *NOTCH1*, *FBXW7*, and *NOTCH3*, Hippo pathway-related gene of *FAT1* and *PTCH1*, and the histone-modifying gene of *KMT2D* (Fig. 3a). The genomic regions of the loss-of-function mutational genes were also frequently loss of copy number in previous ESCC CNV analyses[10,15], which indicated their tumor-suppressing functions in ESCC. We also collected 14 genes whose mutations had recommended

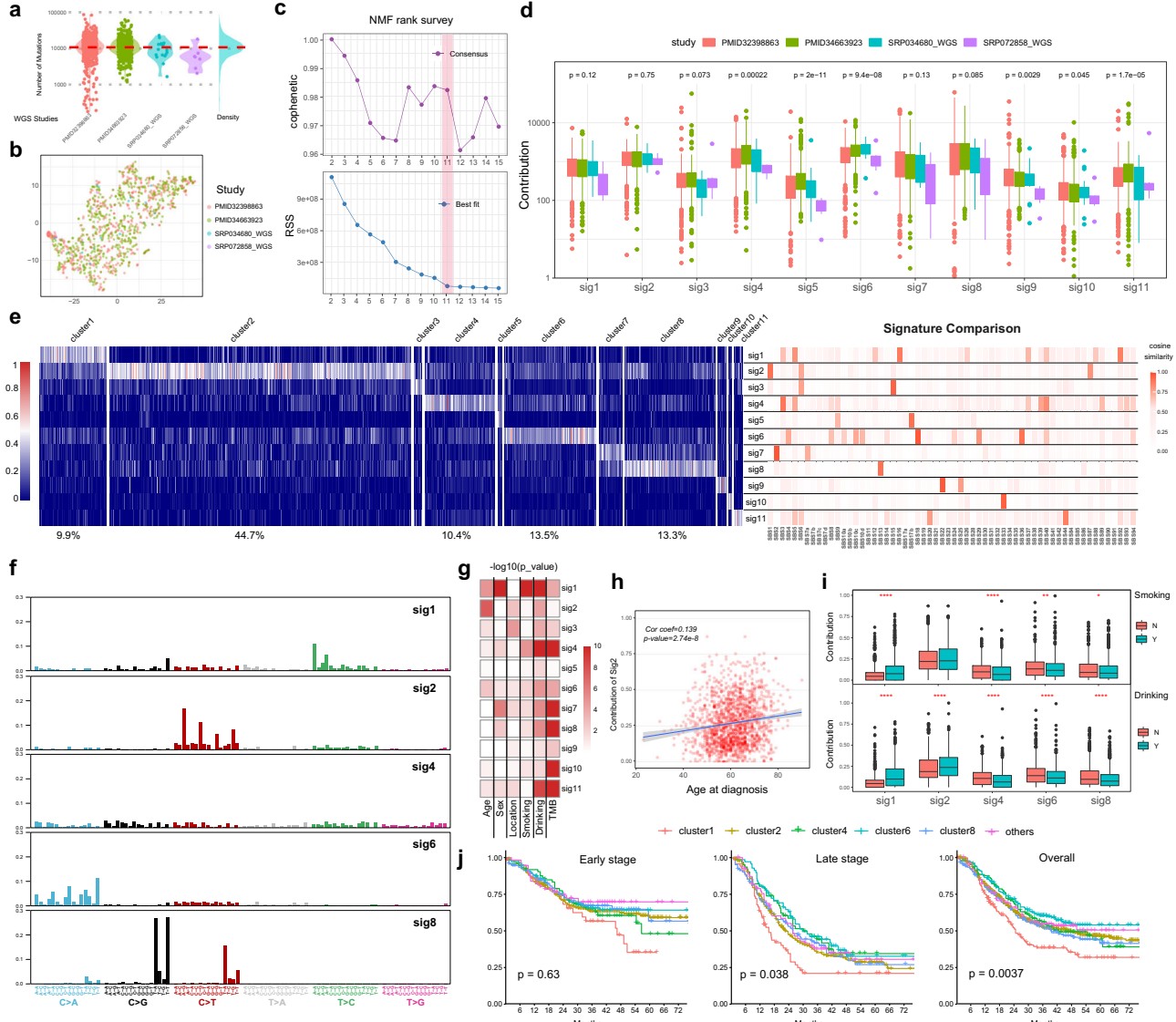

**Fig. 2 | Mutational signature analysis. a** The distribution of total somatic SNVs in the WGS genomes from four datasets. **b** The results of the t-SNE analysis. The count matrix of 96 mutational types in WGS samples (n = 1084) was used in the t-SNE analysis, and the dots were colored by the source of dataset. **c** The NMF rank survey to choose the best separation. The cophenetic correlation coefficient (upper) and the residual sum of squares (lower) were plotted against factorization ranks (from 2 to 15). **d** The contributions of 11 identified signatures in WGS genomes (discovery set, 1084 patients). **e** The contributions of the identified 11 signatures in all ESCC-META genome. In the left panel, the patients were ranked according to their major signatures and grouped to 11 clusters. The right panel laid the heatmap of cosine similarity of the 11 signatures to the COSMIC database. **f** The 96 mutational type features of the sig1, sig2, sig4, sig6, and sig8, which are major mutational signatures in ESCC. **g** The heatmap of the significance (−log10pvalue) of association between signature contributions and the clinical variables in ESCC-META cohort. The two-side Krustal−Wallis test was used to test the difference among clinical groups. **h** The

contribution of sig2 against the age of diagnosis in ESCC-META cohort. The Pearson's correlation coefficient and its significance test were used to measure the correlation. The blue line and the gray band represent the fitted regression line and 95% confidence intervals. **i** In the patients of ESCC-META cohort with available smoking or drinking record, the contributions of major signatures among smoking (upper, n = 1578) or drinking (lower, n = 1484) status. **j** The overall survival curve of the major clusters in early (n = 607) or late-stage patients (n = 639). The labeled p-values were calculated by log-rank test. In **d**, **g**, **h**, and **i**, * indicates $p < 0.05$, **$p < 0.01$, ***$p < 0.001$, ****$p < 0.0001$. In boxplots of **d** and **i**, the lower extreme line, lower end of box, inner line of box, upper end of box, and upper extreme line represent the value of (Q1 − 1.5×IQR), Q1, Q2, Q3 and (Q3 + 1.5×IQR), respectively. Q1−25th quartile; Q2−50th quartile or the median value; Q3−75th quartile. The interquartile range (IQR) is distance between Q1 and Q3 (Q3 − Q1). Source data are provided as a Source Data file.

target drugs at present (Supplementary Table 2), and conceptually defined the nonsilent mutations among the 14 genes as druggable mutations. We could see that 14.7% patients carried somatic mutations in at least one druggable gene, such as *BRCA1/2* (5%) *ROS1*(2%), *EGFR* (2%), and *KRAS* (1%) (Fig. 3a).

More than 27% ESCC patients had somatic mutations in DNA-repair pathway genes, including *BRCA2* (3%), *TDG* (3%), *FANCM* (3%), *RIF* (3%), and *ATM* (3%). Tumors with one or more somatic mutations in these genes present significantly higher mutational load (Fig. 3b, c) and

higher mutational signature contributions of sig7 and sig8 (APOBEC-related process, Supplementary Fig. 4) compared with wild-type tumors. These findings suggested interaction or synergy between APOBEC-associated mutagenesis and somatic altered DNA-repair pathway.

### Significantly mutated genes and mutational hotspots
In the ESCC-META cohort, total 1888 genes mutated in more than 1% patients (Supplementary Data 7), and the top 100 common genes were

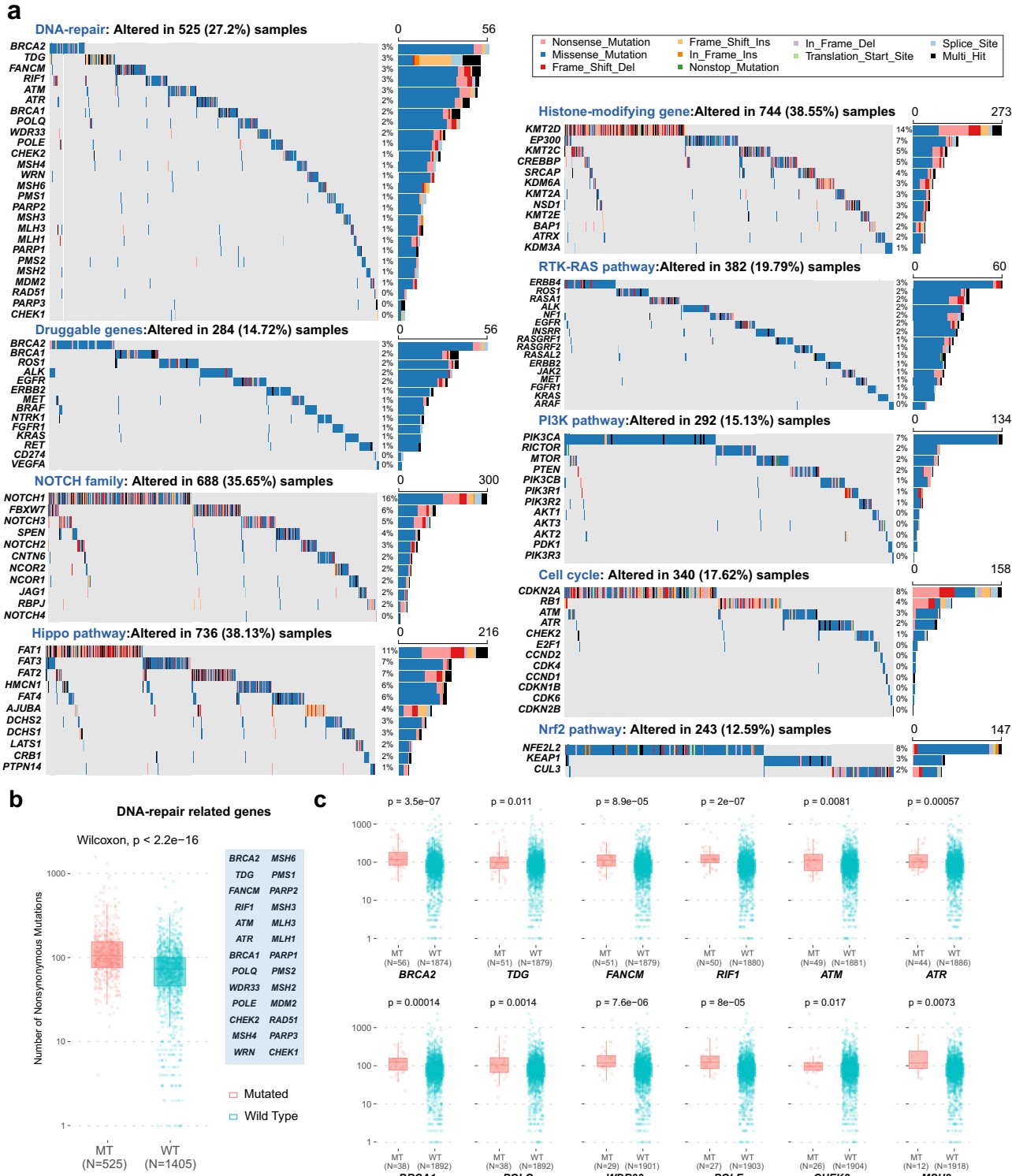

**Fig. 3 | Summary of altered pathways in ESCC-META. a** The oncoplots of genes in mainly altered pathways. The text above each oncoplot indicates the cumulative altered frequencies among ESCC-META cohort, and the right bar plot indicates the number of mutated patients for each gene. The Multi-Hit (black color) represents two or more nonsilent mutational sites of the specified gene in one patient. **b, c** Comparison of mutational load between mutational status of DNA-repair pathway-related genes in ESCC-META cohort. The two-side the Wilcoxon test was used to estimate the significance between two groups. In the boxplots (**b, c**), the lower extreme line, lower end of box, inner line of box, upper end of box, and upper extreme line represent the value of (Q1 − 1.5×IQR), Q1, Q2, Q3 and (Q3 + 1.5×IQR), respectively. Q1−25th quartile; Q2−50th quartile or the median value; Q3−75th quartile. The interquartile range (IQR) is distance between Q1 and Q3 (Q3 − Q1). The effects of single mutated gene are shown in **c** and the effect of any mutation in the pathway (genes listed in the blue box) is shown in **b**. Source data are provided as a Source Data file.

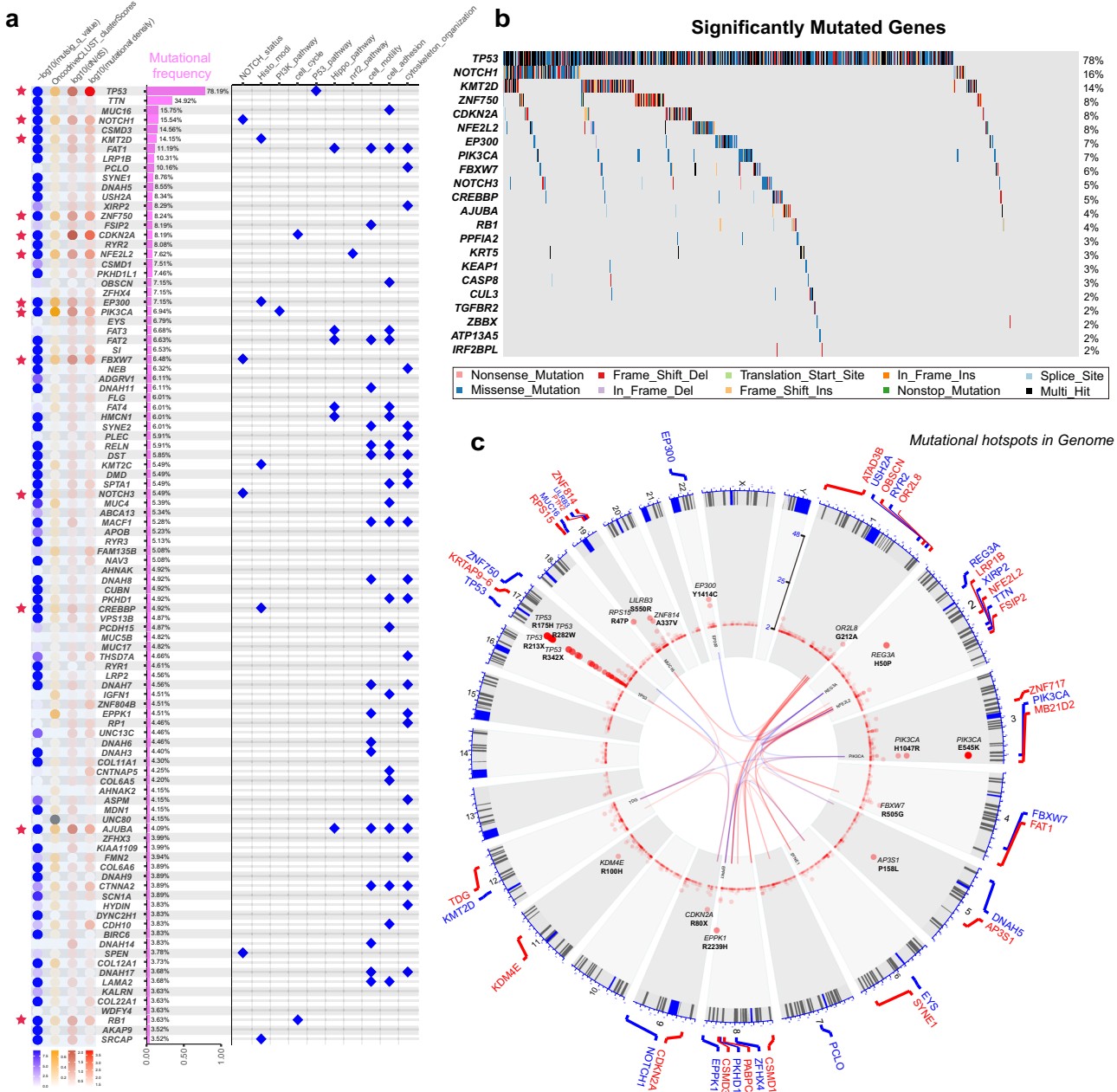

**Fig. 4 | Significantly mutated genes. a** Summary of the top 100 common mutated genes in the ESCC-META dataset. The dot heat of left panel indicated the mutational importance estimated by four approaches, and the red stars labeled the significantly mutated genes (*n* = 22) in combined selection. The bar plot of middle panel indicates the mutational frequency. The dot plot of right panel indicates the major related pathway. **b** The oncoplot of the 22 significantly mutated genes in ESCC-META cohort. **c** The circle chart to indicate the recurrent SNVs in ESCC-META cohort. Each point represented one recurrent mutational site in genome, and the relative height indicate the recurrent frequency. The inner part of the circle linked the significant interactions of gene-pairs, in which the blue links indicated mutually exclusive patterns and the red links indicated co-occurring patterns. Source data are provided as a Source Data file.

summarized in Fig. 3a. To identify the most important mutated genes in tumorigenesis, we applied combined methods to evaluate the significance of the 1888 genes. This strategy selected the most important mutational genes by four aspects to avoid the limitation of a single approach (see Methods for detail) and identified 22 significant genes (Fig. 4b, Supplementary Table 3). Apart from previously reported genes, the *PPFIA2, TGFBR2, ZBBX, ATP13A5,* and *IRF2BPL* were firstly identified as significantly mutated genes in ESCC.

On the other side, the ESCC-META dataset totally included 179,531 unique nonsilent mutational sites distributed in 18097 genes, and only 6917 of them (3.9%) could be detected in two or more patients. We visualized the genome-wide mutational hotspots in Fig. 4c, which

indicated prominent mutational hotspots in chromosome 1q (*OR2L8*), 2(*REG3A, NFE2L2*), 3q (*PIK3CA*), 4q(*FBXW7*), 8q(*EPPK1*), 9p(*CDKN2A*), 17p(*TP53*), 19(*RPS15, LILRB3, ZNF814*), and 22q (*EP3OO*).

As expected, we could detect many mutational hotspots in *TP53*, among which many were located in methylated CpG sites and originally encoded conserved arginine residues[29], such as R175H, R213X, R273H, R248Q, R282W, and R342X. Other previously identified hotspots in ESCC, including the silent mutations of R80X, R58X, and W110X in *CDKN2A*, E545K, and H1047R in *PIK3CA*[30], and the mutations in KEAP1 binding motifs of *NFE2L2*[31], were also confirmed in ESCC-META dataset (Fig. 5a). We detected recurrent frameshift deletions in *TGFBR2* (c.374delA) and *IRF2BPL* (c.224_305del and c.225_303del) in

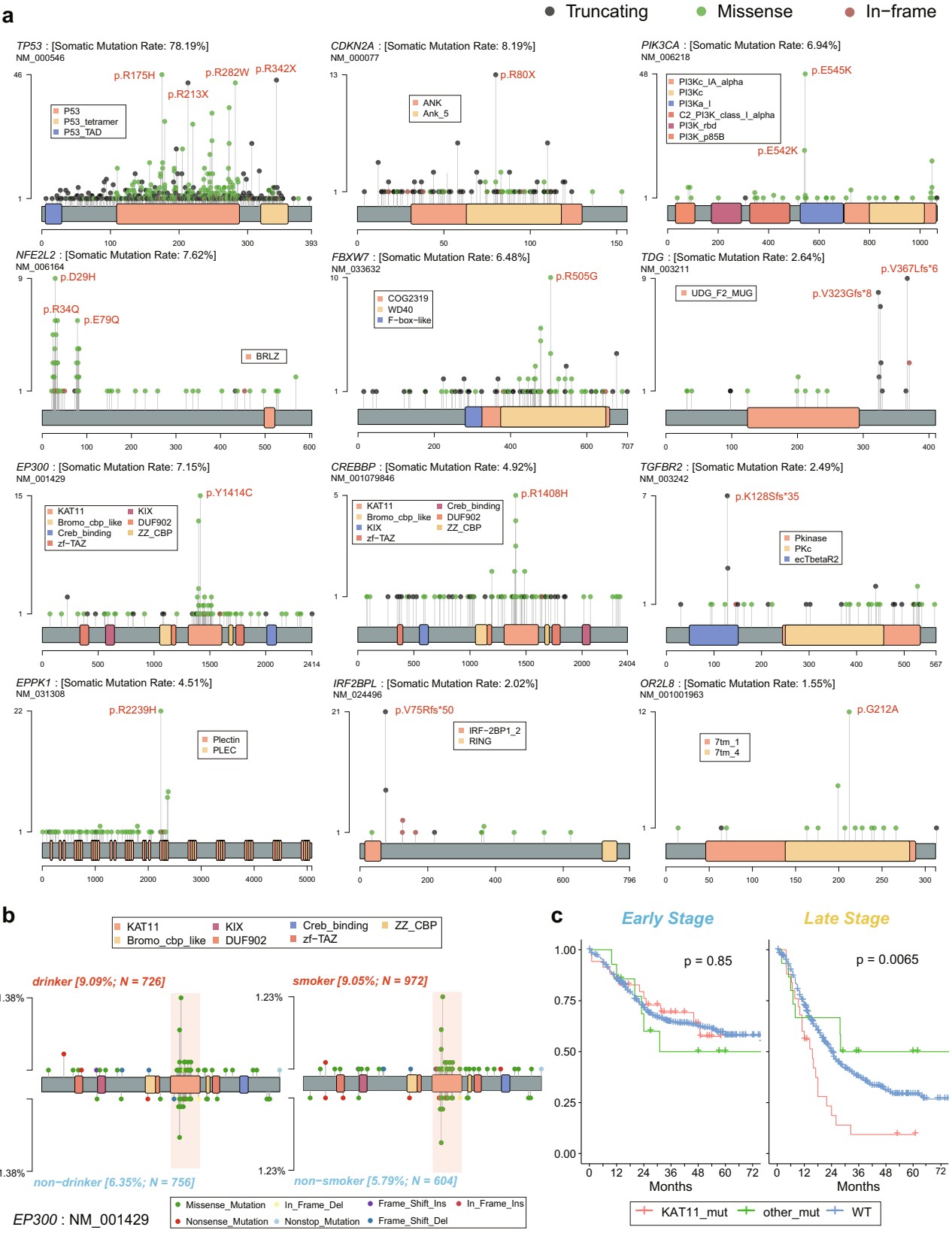

**Fig. 5 | The distribution of mutational hotspots. a** The lollipop plots of some mutational hotspots in the ESCC-META dataset. **b** The comparative lollipop plots of *EP300* in the comparison of drinking (left) or smoking (right) status. The range of KAT11 domain is marked by pink band. **c** The survival comparison between different *EP300* mutational status in early (left panel) or late (right panel) patients of ESCC-META cohort. The two-side log-rank tests were used to indicate significance. Source data are provided as a Source Data file.

ESCC, which were verified in different studies located near the N-terminal of the coding region, indicated their tumor-suppressing functions in ESCC. The *TGFBR2* gene played an important role in TGF-beta pathway, and had been well studied in colon cancer[32]. The *IRF2BPL* encoded an E3 ubiquitin protein ligase and could regulate Wnt signaling pathway in gastric cancer[33]. The two genes were also among the 22 most significant mutated genes identified by multiple approaches (Fig. 4b).

Some coding regions of *EPPK1*, *OR2L8* were not covered in some WES platforms; their total mutational frequencies in the integrated dataset might be slightly underestimated. Nevertheless, we could still find the mutational preference in their coding region. The *EPPK1* encoding protein of Epiplakin contained 13 tandem plakin repeat domains (PRD) and participated in the organization of the cytoskeleton and adhesion complexes[34]. Interestingly, all of the identified non-silent SNVs in *EPPK1* were only located within the first half coding region (from first to the eighth PRD), among which 48.3% mutations occurred in the eighth PRD, including the hotspot site of R2239H (Fig. 5a). This mutational pattern was also observed in other tumor types according to the COSMIC Cancer Gene Census database, but could not be well explained currently.

The histone modification gene of *EP300* and its paralogous gene of *CREBBP* both presented enriched mutational points in the KAT11 domain, which was required for histone acetylation. The mutations in KAT11 of *EP300* were more common in smoking and drinking patients (Fig. 5b), and associated with worse prognosis in late-stage ESCC patients compared with mutations in other *EP300* regions or wild types (Fig. 5c). The mutational interaction analysis indicated mutually exclusive patterns in *EP300* to *CDKN2A* (OR = 0.58) and *EP300* to *NFE2L2* (OR = 0.34), which was unusual in view of the dominant co-occurring patterns for most gene-pairs (the inner part of Fig. 4c, Supplementary Fig. 5 and Supplementary Data 8), which collectively suggested its specific oncogenic functions in ESCC.

## Mutations related to clinical characteristics

In the previous analysis of mutational signatures, we had identified the age-related (sig2 or SBS1) and drinking or smoking-related signature (Fig. 2e). In the ESCC-META cohort, the majority of patients were diagnosed during the age from 50 to 70, whereas 210 patients (11.4%) were younger than 50 years (as a young group) and other 242 patients (12.5%) were older than 70 years (as old group). The mutational frequencies of *NOTCH1*, *XIRP2*, and *NOTCH3* were significantly higher in old group. Notably, the percentage of *NOTCH1* alterations was steadily accumulated with increased diagnostic age of ESCC (6.2%, 10.8%, 12.4%, 15.9%, 20.2%, and 27.3% in groups of ≤40 years, 41–50 years, 51–60 years, 61–70 years, 71–80 years, and a 80 years, respectively, $p < 0.001$ Fig. 6b). The young patients presented more common mutated *PKHD1L1* and *RB1* (Fig. 6a, Supplementary Data 9).

Whereas the esophagus is a long narrow tubular organ from the cervical part to cardia, the tumors from different longitudinal origins might present different genomic profiles. The tumors from the upper thoracic part presented a higher mutational load compared to the middle or lower thoracic part (Supplementary Fig. 6a), while the upper tumor did not have more contribution of APOBEC-related signatures (sig7 and sig8) that were strongly related to the mutational load (Supplementary Fig. 6b). Compare with tumors of the upper or middle thoracic part, the lower tumors had less contribution of sig6 (match to SBS18) that related to damage by reactive oxygen species, and correspondingly, the upper tumor compared to the lower tumor had a significantly higher mutational frequency of *NFE2L2*, which was responsible for antioxidant response and always had gain-of-function mutations (Fig. 6c, Supplementary Fig. 6b).

We noticed that ESCC tumors of the upper or lower thoracic part presented different mutational proneness in other genes. The mutational frequencies of *TEP1*, *DMXL1,* and *NOS1* were higher in the upper thoracic part, while the mutations of *MUC16*, *NOTCH1* were more common in the lower thoracic part (Fig. 6c, Supplementary Data 9). The enrichment analysis of the top different genes showed that the upper part prone genes were enriched to the cytoskeleton organization pathway, while the lower-part prone genes were related to Notch signaling pathway (Fig. 6d).

We next systematically identify prognostic genes by log-rank test and multivariable-adjusted Cox analysis. We distinguished early-stage (stage I or II) patients and late-stage (stage III or IV) patients in subsequent survival analysis because of the significantly differed prognosis and divergent mutational load (as previously indicated) in different tumor stages. We detected some genes whose mutational status was associated with worse prognosis in both early and late-stage patients, such as *PRUNE2*, *TMEM132C*, and *NRXN1* (Supplementary Fig. 6c).

However, some genes presented inconsistent prognostic effects between early and late-stage tumors (Fig. 6e), including the mutations in KAT11 domain of *EP300* that suggested a bad prognosis only in late-stage patients (Fig. 6c). The mutations in *CDKN2A*, *LAMA3*, and *NALCN* were associated with better survival in early patients, but not in late patients, while mutations in *NFE2L2*, *FBN2*, *RNF213*, and *ATP10D* related to bad prognosis in late-stage patients, but not in early-stage patients. The mutational status of *PIK3CA* related to worse prognosis in the early stage but with better prognosis in the late-stage, although there was no significant varied frequency or distributions of mutational sites between tumor stages. The prognostic effect of *PIK3CA* mutations was controversial in previous reports[35–39], and our results of the tumor stage-related prognostic effect might be a possible reason for the discrepant reports. The different influence of mutational status in these genes might be caused by the varied importance of their role in different tumor stages. For example, the mutations of *NFE2L2* were mainly gain-of-function and had been proved to increase the drug or radiation resistance in ESCC[40,41]. This alteration could bring a significantly bad impact on late-stage patients whose major anti-tumor therapies were chemoradiotherapy, but less influence on early-stage patients, for whom the radical surgery played more an important therapeutic role.

## The mutational score could predict the prognosis

Although we identified many independent prognostic genes in the ESCC-META cohort, most of them mutated in less than 5% ESCC patients, which limited the direct application because of the low positive rate. Here we proposed the concept of the mutational score as a combined prognostic model for ESCC. Briefly, based on a large genomic cohort as a discovery set, we firstly selected the candidate prognostic genes by multivariable-adjusted Cox regression, and then combined the top genes as a panel to predict survival outcome (see Methods for details). The mutational score was defined as the count of total somatic nonsynonymous mutated genes in the panel. We set our own sequence dataset of ECRT ($n = 42$) as a testing set, and the rest of ESCC-META cohort with valid survival information ($n = 1476$) as a discovery set for gene selection.

We balanced the positive rate and the complexity of the model to decide the optimal number of genes (see Methods). The final selected eight genes were *NFE2L2*, *CSMD1*, *CREBBP*, *KALRN*, *PRUNE2*, *NRXN1*, *AKAP9,* and *FREM2* (Fig. 7a), among which five genes (*NFE2L2*[42], *CSMD1*[43], *CREBBP*[44], *PRUNE2*[45], and *AKAP9*[46]) had been reported as tumor-suppressing gene or with dominant gain-of-function mutational hotspots in tumors. The sum of nonsilent mutations in the eight-gene panel was defined as the mutational score of ESCC. Unsurprisingly, the mutational score showed a positive correlation with total mutational load in ESCC (Fig. 7b). In the discovery set, 29.1% of early-stage patients and 27.8% of late-stage patients could detect at least one nonsilent mutation in eight-gene panels (Fig. 7c). In early-stage patients, one positive gene in mutational score panel implied 1.78 of HR compared

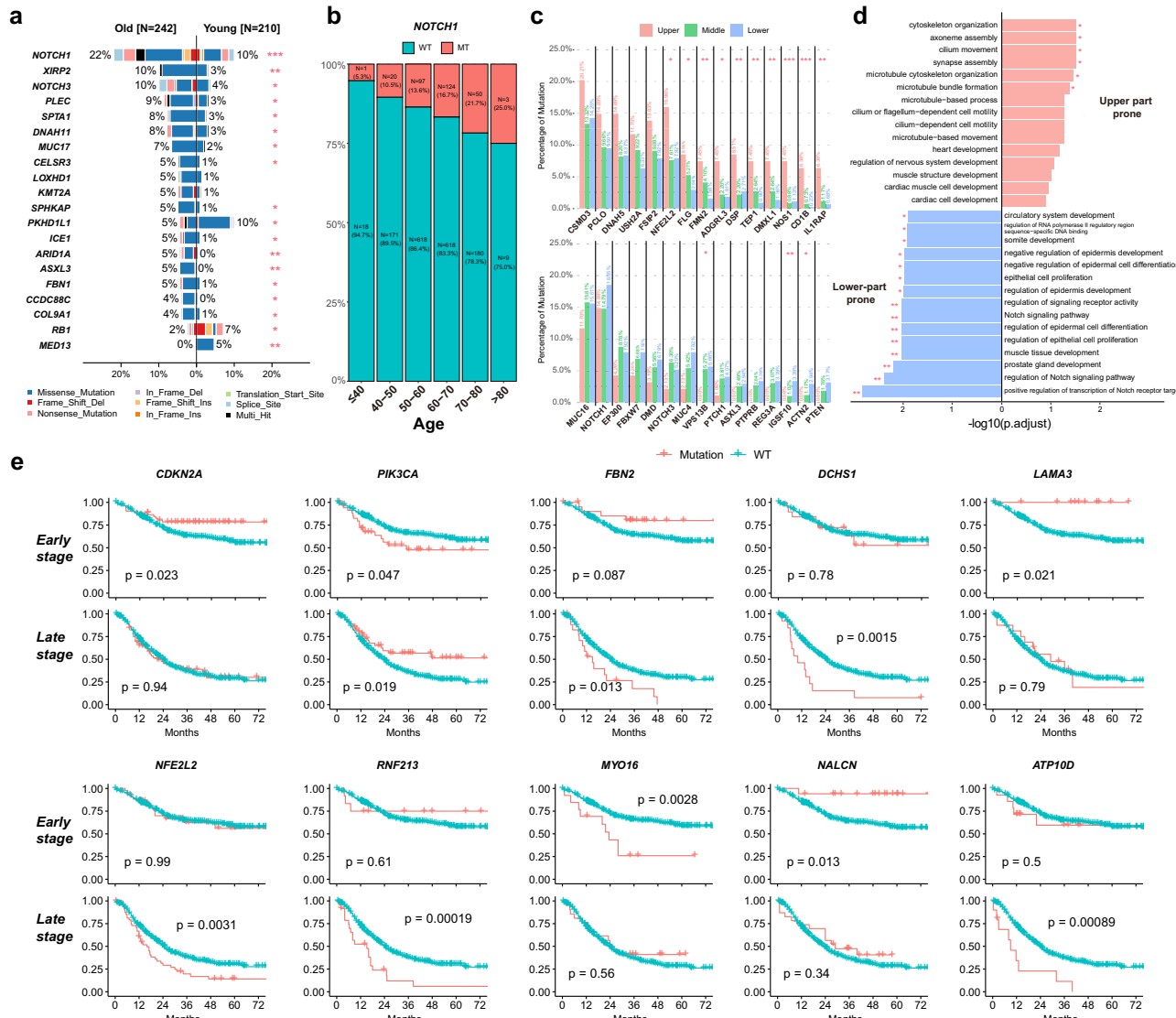

**Fig. 6 | Clinical characteristics related to genomic features. a** Comparative bar plot of most significantly varied genes between old patients and young patients. The two-side Fisher's exact test was used to indicate the significance, and * indicating *p* < 0.05, **\*\****p* < 0.01, \*\*\**p* < 0.001. **b** The proportion of *NOTCH1* mutated patients in different groups of diagnostic age. **c** The mutational frequencies in tumors from different thoracic part. The upper panel indicated genes more commonly mutated in upper part, while the lower panel presented lower-part prone mutations. The two-side Fisher's exact test was used to indicate the significance, and * indicating *p* < 0.05, **\*\****p* < 0.01, \*\*\**p* < 0.001. **d** The top 15 enriched pathways from GO analysis of upper part prone genes (upper part) or lower-part prone genes (lower part). The labeled * represents for *p* (adjusted) <0.05, ** for *p* (adjusted) <0.01. **e** Survival plots of some significant genes in early or late-stage patients. The two-side log-rank test was used to indicate the significance. Source data are provided as a Source Data file.

to negative patients, and two or more positive genes suggested 2.26 of HR value. For late-stage patients, the one gene mutation and two or more mutations indicated 1.49 and 2.28 HR, respectively (Fig. 7d). We further evaluated its prognostic value in separated datasets by stage-adjusted HR in Cox regression. In the 13 single datasets that included at least 30 patients with survival information, the adjusted HR values indicated a similar trend of worse survival in positive patients (HR > 1, Fig. 7e), which suggested its prognostic value was generally effective in the discovery set without systematic bias of data sources.

We next verified the mutational score in an independent testing set of ECRT, which involved patients from a phase III ESCC clinical trial (Fig. 7f, see Methods for details). In the testing set, one mutated gene presented 2.99 HR, and two all more mutations indicated 4.93 HR (log-rank *p*-value = 0.028, Fig. 7g), which verified the mutational score as an effective predictor of bad prognosis in ESCC. We also performed multivariable Cox regression to recognize potential confounding of clinical variables or mutational signatures, which proved mutational

score as an independent prognostic predictor. Compared with 0 value of the score, one mutation and two or more mutations implied multivariable-adjusted HR [95% CI] values of 1.53 [1.29–1.8] and 2.17 [1.63–2.9] (Supplementary Fig. 7). Collectively, the eight-gene mutational score was validated as a robust prognostic model in ESCC for clinical application.

## Discussion
Based on publicly available datasets and our own sequencing results, we presented the ESCC mutational landscape of the ESCC-META. The integrated work combining dozens of studies could well utilize previously published genomic resources, provide updated results, and obtain more trustworthy evidence in this field.

We had made many efforts to evaluate and reduce the heterogeneity among data sources. As we had proved in the results, using a set of quality control and integration processes, we could obtain well-homogenized SNV records in the coding region. However, the batch

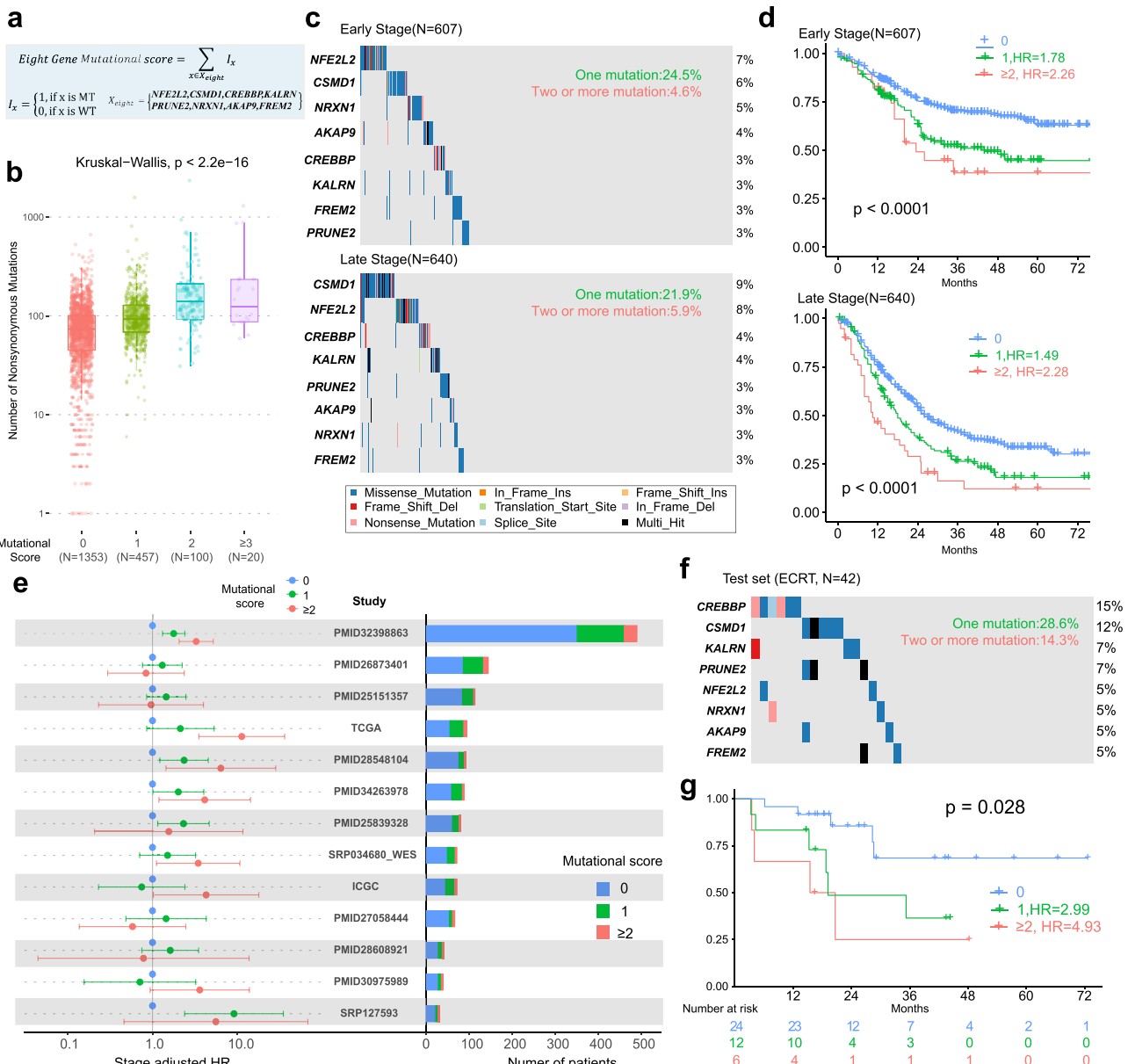

**Fig. 7 | Building of eight-gene mutational score. a** The formula definition of the eight-gene mutational score. WT wide type, MT mutation. **b** The comparison of mutational load among different mutation score in all ESCC-META genomes, the two-side Krustal–Wallis test was used to estimate the significance among the groups. In the boxplots, the lower extreme line, lower end of box, inner line of box, upper end of box, and upper extreme line represent the value of (Q1 − 1.5×IQR), Q1, Q2, Q3 and (Q3 + 1.5×IQR), respectively. Q1−25th quartile; Q2−50th quartile or the median value; Q3−75th quartile. The interquartile range (IQR) is distance between Q1 and Q3 (Q3 − Q1). **c** Oncoplots of the eight genes in mutational score within early-stage patients (upper) or late-stage patients (lower) of discovery set. **d** The survival comparison between different mutational scores within early-stage patients (upper, *n* = 607) or late-stage patients (lower, *n* = 640) of discovery set. The two-side log-rank test was used to indicate the significance. **e** The prognostic value of mutational score within separated dataset. The left panel indicates the stage-adjusted HR of mutational score with the 95% confidence interval (the dot and error bar). The left panel indicates the total and positive number in each dataset. **f** The oncoplot of the eight genes in mutational score within test set of ECRT (*n* = 42). **g** The survival comparison between different mutational scores within test set. The two-side log-rank test was used to indicate the significance. Source data are provided as a Source Data file.

effects could not be ignored in copy number variants (CNV) and structure variants (SV). Unlike the accuracy and reproducibility in point mutations, the detection of large-scale variants might be less robust in the current NGS platform, which was more sensitive to the capture platform, sequence mode and data quality, the coverage regions, filter parameters, and calling algorithms. We excluded these records from the current ESCC-META dataset to avoid unreliable analyses.

The ESCC mutational profiles featured moderate or low mutational burdens and high heterogeneity among patients. Although previous studies have reported the prognostic significance of higher

mutational load in ESCC[15,47], in the ESCC-META cohort, we found the mutational load significantly related to tumor stage but did not influence overall survival in stage-adjusted analysis. Our results showed that the activity of APOBEC enzymes (sig8, SBS13) partly explained the mutational load in ESCC, and the somatic mutations in DNA repair related could significantly increase the mutational load. Another identified mutational signature (sig4) presented similarity to SBS3, which was associated with homologous recombination deficiency (HRD)[48] and its prevalence was also reported in other ESCC studies[24], but we could not identify its positive relationship with the total mutational load. The lack of reliable large-scale genomic variants, such

as LOH and large insertions and deletions that were more important features of HRD, limited our assessment of the accurate contribution of somatic HRD in ESCC genome.

The development of ESCC was thought to be the result of long-term accumulation of somatic mutation in normal esophageal epithelia[49,50]. We revealed the mutational difference between upper thoracic ESCC and lower thoracic tumors, which indicated the profound influence of tissue microenvironment on oncogenesis. Additionally, our results indicated the age-related alterations in mutational signature and mutational frequency profiles, including the significantly increasing mutational frequency of *NOTCH1*. Our results provided oncogenic evidence on the previously reported findings of age-related somatic mutations in normal tissues, including the prominent marker of *NOTCH1*[51,52].

The initial purpose of building the integrated ESCC-META dataset was to provide supporting data for NGS panel design. Compared with tests of other omics (such as transcriptome), the mutational panel test had advantages in sensitivity, accuracy, and fewer restrictions in sample preparation (not requiring fresh tissue). However, the ESCC genome often presented high heterogeneity and low mutational load. These inherent genomic features implied that most of the significantly mutated genes could only be detected in a small proportion of ESCC patients, and made it difficult to design an effective NGS test panel. We proposed the concept of mutational score that combined multiple significant genes as a test panel to increase the positive proportion in a clinical test. This model was specifically designed for ESCC, and its building was based on a large integrated cohort. Compared with previously reported prediction models, which were often theoretical or platform-dependent, our work had the advantages of robustness and practicality. Owing to the limited involved genes and simplicity of its algorithm, the capture probes for the eight-gene mutational score were also applicable for low abundance DNA libraries, such as circulating tumor DNA (ctDNA) sequence in ESCC. Since the mutational score could distinguish the patients with worse prognosis, its dynamic monitoring in ctDNA would be helpful in individualized treatment.

Our study had other limitations. The ESCC-META dataset did not include germline mutations at present, which also contributed to the tumorigenesis in some ESCC patients[53,54]. The present dataset lacks more details of treatment-related information, which limited more specific discoveries. The application of new treatment regimens such as immunotherapy in ESCC might change the prognostic effects of some mutations. Our team will keep tracing the latest available data and update the integrated dataset to facilitate research in this field.

## Methods

### Data selection
The study was approved by the ethics committee of Shandong Cancer Hospital and Institute, and written informed consent was obtained from all our patients (the ECRT cohort). We hope to collect all public whole-genome sequence (WGS) or whole-exome sequence (WES) data of ESCC. The genomic data were collected from the following three sources.

Firstly, the genomic databases were searched, including NCBI-SRA, EBI-ENA, and NGDC-GSA, for all publicly available raw sequence data. Second, the mutational records in the published article. We search all potential articles in PubMed, and the references of relevant articles were also scanned. The available mutational list should at least include all somatic nonsilent SNVs records for each individual. If the raw sequence data were also available, we directly included and reanalysis their raw sequence data, ignoring their published results. Third, the public cancer genome databases, including TCGA, ICGC-Esophagus, and COSMIC Mutation database. If the cohort was both involved in the published articles and genome databases, we compared them and used the one with more detailed records.

Target sequences other than WES and the low coverage WGS data (mean coverage < 10) for CNV analysis were not included in ESCC-META. If the multiple tumor samples were collected from different time points, only the earliest tumor sample (at diagnosis or before any treatment) was used. We excluded patients with multiple primary tumors or esophageal tumors of unclear pathological diagnosis. The samples apart from primary tumor tissue, such as from metastatic sites were also excluded.

The patient ID was renamed by pasting their source and original sample name. We also processed and checked the available clinical information of each individual, including age, sex, drinking and smoking history, tumor stage, tumor location, and tumor grade. All misleading or vague records were regarded as not available (N.A.).

In our work, we separated the dataset of SRP034680 into SRP072858_WGS (data of WGS part) and SRP072858_WES (data of WES part), the SRP072858 into SRP072858_WGS (data of WGS part) and SRP072858_WES (data of WES part), the SRP099292 into SRP099292_S (single tumor sample per patient), and SRP099292_M (multiple tumor samples per patient).

### Sequencing of ECRT dataset
The ECRT dataset was sequenced from patients involved in a multi-center, randomized phase III clinical trial of ChiCTR-IPR-15007172, which was started in 2015 and approved by the ethics committee of Shandong Cancer Hospital and Institute. Briefly, the patients were all diagnosed with locally advanced ESCC tumors and received radical concurrent chemoradiotherapy as the first tumor treatment. Written informed consent was obtained from all patients of the ECRT cohort. Total 42 patients were included in this ECRT cohort, and the rest patients in the clinical trial were excluded mainly because of no available FFPE tumor tissues or no sufficient DNA extracted for WES sequence.

The formalin fixation and paraffin embedding (FFPE) endoscopic biopsy tumor samples before any treatment were collected. The suitable FFPE samples for sequence must contain more than 50% tumor region under the microscope and have 100 ng available DNA after extraction. The peripheral blood cells of each patient were used as normal control. The genomic DNA was extracted from FFPE by Gene-Read DNA FFPE Kit (QIAGEN) and from peripheral blood cells by PureLink™ Genomic DNA Mini Kit (ThermoFisher). The genomic DNA was fragmented and captured by Agilent SureSelect Human All Exon V6 Kit (Agilent Technologies). The sequencing in PE150 mode was performed in Illumina Novaseq 6000 platform. The least mean coverage of captured region must be more than 100× for the control sample and more than 200× for the tumor sample.

### Processing raw sequence data
If the reads data were NCBI-SRA format, it was converted to fastq file by SRA-Tools (v2.11). The fastq files were firstly performed quality control by fastp (v0.23)[55] with default parameters. The files of different sequence lanes from the same library or the different SRA reads files from the same sample were combined before mapping. The mapping process was performed by BWA (v0.7.1) to hg38.p13 genome. The bam files were then deduplicated and applied base quality score recalibration by GATK (v4.1) according to the recommended practice. The pairwise relationships between tumor and normal samples were examined by BAM-matcher[56], and the mismatched samples were removed. The single nucleotide variants (SNVs) and insertion or deletion mutations (INDELs) were called by Mutect2 in GATK (v4.1). The filter criteria varied within the following three situations.

Firstly, for WES sequence of one tumor sample with normal control, the coverage should be at least 30 in the tumor sample and at least 20 in the normal sample, at least three alternative reads in the tumor sample to support the variant call, and mutation frequency at least 0.05. Second, for WGS sequence of one tumor sample with normal

control, the coverage should be at least 20 in the tumor sample and at least 20 in the normal sample; the rest is the same as above. Third, for WES sequence of multiple tumor samples with one normal control from the same patient, the included variant should be detected in at least one tumor sample meeting the above criteria, additionally, the variant base should be identified (at least two alternative support reads) in another tumor sample.

### Preparation of mutational records

For mutational records from the reported lists (such as in MAF format) or databases without raw reads data, the authenticity was firstly checked by base comparison. For example, if the raw mutational record is chr19:63554635, G > T in hg18, the reference base in hg18 of chr19:63554635 should be G, if not, this record was suspicious and must be re-examined. The verified records from each dataset were then prepared to VCF format (Version 4.2) and transformed to hg38 by CrossMap (v0.2.6)[57], which will also remove a few records because of failure to convert. The converted mutational lists were recheck with hg38 as the first step. The number of raw and verified SNVs was listed in Supplementary Data 1.

### Integration and annotation

The involved patients and their genomes were firstly renamed by pasting their source dataset and their original names. The duplicated samples were carefully identified by checking their source information and pairwise comparisons of the mutational profiles. We only keep one original sample and exclude all duplicated data in the final integration.

The quality-controlled results were then combined into a single VCF file and annotated by ANNOVAR (December 2019 version)[58]. This combined VCF file including all filtered mutational records (including mutations in noncoding regions) and was used in the mutational signature analysis, while we only used the nonsilent mutational records according to the annotation results for the rest analysis.

### Comparison between capture platforms

The integrated dataset included genomes from WES of different capture platforms (Supplementary Data 1 and Supplementary Table 1). Two studies used Agilent SureSelect V4, eight studies used Agilent SureSelect V5, four studies used Agilent SureSelect V6, two used NimbleGen SeqCap EZ Exome, one used Agilent SureSelect Clinical, and one used Agilent TruSeq Exome. The capture platforms of the rest 7 WES studies were unable to be identified. The capture region files of Agilent SureSelect V4, V5, V6, and Agilent SureSelect Clinical were downloaded from the website of Agilent Technologies, and the rest two platforms were downloaded from UCSC database. All the region files were transformed to hg38 by CrossMap.

The ESCC WGS genomes from four studies (PMID28548104, PMID32398863, SRP072858_WGS, and SRP034680_WGS) totally included 55,980 nonsynonymous SNVs and were set as a test set. The genomes sequenced by the capture platform were also examined as a reference set to estimate background distribution. For each SNVs, we calculated the distance between its locus to the nearest capture boundary. The positive value represented of capture range of the mutational site, and the negative value represented the capture range.

The distribution of the distance was shown in Supplementary Fig. 2a. We could see that some SNVs were detected within the flank regions in both the reference set and the test set. We noticed that even in the reference set, there did exist reported SNVs far away from the capture region, especially for Agilent SureSelect V4 platform. It was partly because some regions in the original capture files of hg19 failed to convert into hg38. Consequently, we thought that the percentage of SNVs located more than 200 bp distance in the test set subtracted from the percentage in the reference should be a reasonable estimation of the influence of the capture platform. The detailed results were visualized in Supplementary Fig. 2. Although the different capture platforms led to less than 1% nonsilent detected SNVs in ESCC, the uncovered coding regions of some genes could induce bias in integrated analysis. The genes were labeled in Supplementary Fig. 2b and the regions were listed in Supplementary Data 3.

### Identification of significantly mutated genes

Genomes from three studies (PMID22877736, $n = 12$; PMID32929369, $n = 14$; PMID28608921, $n = 41$) were excluded in this part of the analysis because these records only contained nonsilent mutations and without available synonymous mutations, which might increase the false-positive rate in integrated analysis. The remaining 1863 ESCC genomes were included. Besides mutational frequency, we used the following four approaches to identify the most important genes.

Firstly, we applied MutSigCV to calculate the Q value, which was designed to identify genes that were mutated more often than expected by chance, given background mutation processes[59]. The MAF file and other input files (coverage table, covariates table, and mutation type dictionary file) were prepared to comply with its requirements. Note that the MutSigCV may not produce reliable results on cancers with low mutation frequencies like ESCC due to its internal assumptions[59], thus this part of the results should be interpreted with caution. Second, we applied oncodriveCLUST to calculate the cluster score with default parameters. The oncodriveCLUST was designed to find driver genes with enriched mutational hotspots[60]. Third, we calculated the ratio of nonsynonymous mutation to synonymous substitution (dN/dS) in the coding region (CDS) for each gene. The high dN/dS values suggested positive selection in cancer evolution[61]. Fourthly, we calculated the coding length adjusted mutational frequency for each gene as Eq. (1) and defined it as mutational density in this article.

$$\text{Mutational density} = \frac{\text{N of synonymous mutations}/\text{N of total patients}}{\text{length of CDS(Mb)}}$$

(1)

In the calculations of 2,3,4 approaches, the most common transcript was specified based on the SNVs annotation results for each gene. We applied the following five criteria to obtain the most significant genes: 1, mutational frequency ≥ 2%; 2, MutSigCV Q value ≤ 0.01; 3, OncodriveCLUST clusterScores ≥ 0.2; 4, mutational density ≥ 50; and 5, dN/dS ≥ 5.

### Mutational signature analysis

The mutational signature analysis was performed based on the matrix of 96 types of base substitutions, including the six substitution classes (C > A, C > G, C > T, T > A, T > C, T > G) combined with substitutions in the context of left and right flanking bases. The non-negative matrix factorization (NMF) algorithm[62] was employed to decompose the major k mutational signatures and their contributions to each genome. The optimal number of separations (k) was selected both considering the cophenetic correlations and the residual sum of squares (RSS)[25]. We chose 11 as the best number of separation because the cophenetic correlations presented a maximum decrease between k = 11 and k = 12, and the declines of RSS were obviously slower in higher k value (Fig. 2c).

In the discovery set of WGS sample, the scaled basis components from the NMF model were extracted as the identified mutational signatures. The contribution of these signatures in all ESCC-META samples was subsequently predicted. The COSMIC Mutational Signatures database (v3) was used as a reference for comparison (measured by cosine similarities) and interpretation.

### Building mutational score

The intention of the proposed mutational score is to overcome the applied limitation of prognostic genes whose mutational frequencies

were low. We want to establish a gene model that could be applicable and robust in the real world. It requires a large genomic cohort as a training set, and a simple but reliable algorithm to avoid the risk of overfitting. The genes that were located in incomplete covered regions by one or some capture platforms (Supplementary Fig. 2b, such as *MUC4*, *AP3S1,* and *OR2L8*) were excluded from the process to avoid potential artifacts in survival analysis. Based on the ESCC-META cohort, we use the following four steps to establish the score.

Firstly, we select the patients with overall survival information (survival time and status) as the training set. The multivariable-adjusted Cox analysis was performed for each gene to obtain the hazard ratio (HR) of mutational status (mutated to wild-type, MT to WT). The adjusted variables include age, sex, and tumor stage. Second, the adjusted Cox analyses were also performed in the subgroup of early-stage and late-stage patients of the training set to obtain the stage-specific HR value. These results were presented in Supplementary Data 10. Third, the candidate genes were selected with the following two criteria: (a) the mutational state of the gene was significantly associated with worse survival in the overall training set (adjusted HR overall > 1 and $p < 0.05$), and (b) the trend of association remained in early-stage and late-stage subgroup (adjusted HR > 1 in both early and late stage). Fourth, the candidate genes were ranked by their mutational frequencies, and the top n genes ($X_n$) were included to build the mutational score. The mutational score is defined as the simple sum of the somatic nonsynonymous mutations in this panel of genes (Eq. 2).

$$\text{Mutational score} = \sum_{x \in X_n} I_x; I_x = \begin{cases} 1, \text{if } x \text{ is Mutated} \\ 0, \text{if } x \text{ is Wild type} \end{cases} \quad (2)$$

In ESCC, we used the top eight genes ($n = 8$) to build the mutational score under two considerations. Firstly, we hope this model could distinguish around one-third of patients with a worse prognosis, thus the least number of genes must be included to avoid the low positive rate in the application. Second, due to the high heterogeneity of the ESCC genome, the marginal effect of the increased number of genes above a certain value would be significantly decreased. We noticed that, except for the top eight genes, the mutational frequencies of the rest genes were much lower (no more than 3%) and would contribute little to the total positive rate. Additionally, the more genes selected, the more complexity of the model and the more challenge in its application, therefore we selected the top eight genes and excluded other genes in the panel.

**Statistical analysis**
Linear regression was used to estimate the potential systematic batch effects within datasets. The random effect model in the meta-analysis was employed to estimate the inverse variance weighted pooled mutational frequencies[63]. The dimensionality reduction method of t-SNE was used to indicate the potential batch effects in mutational genes matrix or mutational types matrix. The mutually exclusive or co-occurring genes were evaluated by the odd ratio of their co-mutation in the whole dataset, and tested by the two-sided Fisher's Exact test. The Fisher exact test was also used in other category variable comparisons among groups. The Wilcox test (within two groups) and Kruskal–Wallis test (within multiple groups) were used in grouped continuous variable comparisons. The Kaplan–Meier curves and log-rank tests were performed in survival analyses. Multivariable Cox proportional hazards regression was employed to calculate the adjusted hazard ratio (HR).

The statistical analysis and visualizations were all performed in R (4.1.0) with the help of packages of survival (3.2), survminer (0.4.9), meta (4.9), maftools (2.8), Rtsne (0.15), NMF (0.23.0), mutSignatures (2.1.1), dplyr (1.0.6), and ggplot2 (3.3.5).

## Reporting summary
Further information on research design is available in the Nature Research Reporting Summary linked to this article.

## Data availability
The raw WES data of the ECRT cohort generated in this study have been deposited in National Genomics Data Center (NGDC) of China National Center for Bioinformation under the accession code of HRA002596. The raw sequence data are available under controlled access to avoid misuse of the human genomic data. Requests for academic purposes only will be processed by the Data Access Committee (DAC) via the GSA platform within ~2 weeks. Once access has been granted, the data will be available to download for 3 months. The public raw sequencing data used in this study are available in the SRA database under accession codes SRP099292, SRP033394, SRP059537, SRP150544, SRP072112, SRP179388, SRP072858, SRP127593, SRP327447, SRP034680, and SRP116657. The public level 3 mutational records are available from the TCGA and ICGC-Esophagus databases, and the public known mutational signature profiles are available from the COSMIC database. The mutational records in coding region of the integrated ESCC-META datasets are public available at synapse under the accession code of syn27304838. The SNVs in the noncoding regions and the full clinical information, including the basic characteristic of patients, the diagnosis of tumors, and the survival information, would be provided on request. The remaining data are available within the Article, Supplementary Information, or Source Data file. Source data are provided in this paper.

## Code availability
The custom code we used to establish the ESCC-META dataset is public available in https://github.com/liminghao663/ESCC-META and the corresponding DOI is as follows doi:10.5281/zenodo.6904002[64].

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

## Acknowledgements

This work was supported by the grants of Key Research and Development Program of Shandong Province of China, 2017CXZC1206 (by B.L.), National Natural Science Foundation of China, 81874224 (by B.L.), and Academic promotion program of Shandong First Medical University, China, 2019LJ004 (by B.L.).

## Author contributions

M.L.: Data curation (Lead); Formal analysis (Lead); Investigation (Lead); Methodology (Lead); Software (Lead); Visualization (Lead); Writing—original draft (Lead). Z.Z.: Conceptualization (Equal); Data curation (Equal); Investigation (Equal); Resources (Equal); Validation (Equal); Writing—original draft (Equal); Writing—review and editing (Equal). Q.W.: Data curation (Equal); Resources (Equal); Validation (Equal). Y.Y.: Data curation (Equal); Resources (Equal); Validation (Equal). B.L.: Conceptualization (Lead); Funding acquisition (Lead); Validation (Lead); Writing—review and editing (Lead).

## Competing interests

The authors declare no competing interests.
