## [Peer Review File · Nature Communications]

Integrated cohort of esophageal squamous cell cancer revealed genomic features underlying clinical characteristicsReviewers' Comments:

Reviewer #1:

Remarks to the Author:

Thank you for the opportunity to review the manuscript entitled, "Integrated cohort of esophageal squamous cell cancer revealed genomic features underlying clinical characteristics", for publication in Nature Communications.

ESCC is a common, poor-prognosis histologic subtype of esophageal cancer. ESCC biology is highly heterogenous, with poorly defined clinical subtypes and a lack of prognostic biomarkers, despite extensive whole-exome and -genome studies including several large patient cohorts. Better understanding and classification of the genomic etiologies of ESCC could lead to new targeted therapies or improved clinical management.

In the submitted work, the authors combine and jointly re-analyze 19 previously published ESCC cohorts and report on genomic features, potential neoantigens, and correlates of prognosis. The manuscript is mostly clearly written on a technical level. Sufficient details are included to permit reproducibility.

Major criticisms related to how the data support the proposed title- and abstract-level conclusions are:

FIGURE 1A-B

The critical, central technical issue of the authors' work is the extent to which the 19 previously published studies can be jointly re-analyzed. The concern is confounding due to differences in subject clinicopathologic composition and genomic data generation, from sample collection to variant calling. In particular, a notorious issue is the large degree of variability between different variant calling strategies in terms of criteria needed to consider a variant "real." If both clinicopathologic characteristics and genomic data generation pipelines differ across cohorts, the possibility of systematic bias arises.

Figure 1A shows that the distribution of non-synonymous mutations differs across cohorts, in many cases by a margin that appears likely to be statistically significant. Why is this - differences in subject characteristics, differences in data generation, or both? The null hypothesis is that both contribute. The degree of comparability of the 19 studies needs to be directly quantified by the authors, for instance by multivariate modeling that includes study ID as a variable, to understand that the results of the authors' re-analysis are valid. The t-SNE analysis of 1B does not adequately address this issue.

FIGURE 3A:

The authors perform predicted neoantigen analysis by determining if peptides derived from somatic mutations are likely to bind with high affinity to HLA, which is thought to be a prerequisite for immunogenicity.

The authors' neoantigen analysis is incorrect and must be revised or removed from the manuscript. Peptide binding to HLA depends on HLA type, which must be determined for every subject either from sequencing data, for which accurate tools exist, or via another assay. The authors did not determine HLA types for any subjects. The authors instead used the "most common type I HLA alleles in East Asia." Given the large degree of variability across humans in HLA type, the analysis presented is inaccurate.

FIGURE 6B:

The authors combine mutation status for the eight most frequently mutated genes into a boolean

composite mutation score. The biologic rationale of this is not clear. Such a mutation score is likely to be highly correlated with, and thus a surrogate marker for, overall tumor mutation burden. The prognostic significance of TMB is well-studied.

Reviewer #2:

Remarks to the Author:

The manuscript titled: "Integrated cohort of esophageal squamous cell cancer revealed genomic features underlying clinical characteristics" describes a meta-analysis of the somatic SNV and indel mutations in a cohort of 1616 ESCC samples from 19 studies. Through this meta-analysis the authors aim to determine prognostic mutation features.

There are number of major concerns about this paper which affect the robustness, impact and overall conclusions:

- The novelty of the study is not clear. What added advantage has this meta-analysis shown over previous studies of ESCC?
- The analysis of frequently mutated genes is anecdotal. Approaches to look at significantly mutated genes, as opposed to the frequency of mutations would improve this section. Frequency of mutations within genes can be affected by things such as gene size or hotspots for example the TTN gene is very large and may not be significant. The authors should consider running methods such as Intogen and MutSigCV (there are others also) to determine what are the significantly mutated genes, otherwise the findings are not robust.
- The genomic distribution of mutational hotspots section looks for recurrent genomic positions. For recurrent positions many were present in just 2 samples. It would be helpful to know how many of these recurrent positions were only present in 2 samples and how confident are the authors that these are present by chance or passenger events as opposed to driving events. The approach to look at recurrent loci can also be improved by looking at significantly mutated loci (see previous comment).
- The analysis and results presented lack detail making them hard to interpret. As an example, for the TSNE in Figure 1B there are no axis labels, it is not clear what numeric data is being used in the plot (is it absence presence of mutations in specific genes?) and the legend does not contain sufficient detail.
- The findings from this paper hinge on there being significant mutations and survival differences between groups. However, often the statistical test used is not defined in the text or in the figure legend, the p-values reported are sometimes low, the test may be confounded by other factors such as grade or age, and the size of the groups in some of the analyses are unclear - for example in the survival plots in Figure 2D2.
- The mutational signature work needs more detail. Does the meta-analysis find new signatures related to ESCC? Were all mutations used in the analysis (recommended) or just a subset as indicated from line 382-385. In Figure 4D the authors aim to show the correlation of signatures with age, however this would be clearer as a scatter plot of signature contribution vs age, since it is hard to draw conclusions from the proportion plots.
- For the TMB and neoantigen section in Figure 3E and F, how were the groups of "low" and "high" defined? Why was a TMB of 5 selected - is this clinically relevant? The p-value for neoantigen is very low (Figure 3F) at 0.034, was this also correct for other clinical factors such as age, or cohort?
- The prognostic signature based on mutations in 8 genes, could this also be cofounded by other clinical features? How clinically useful would this signature be? What number of samples harbor mutations of these genes? It would be good to validate this 8 gene signature in another cohort, since it is a key finding in the paper.

Minor comments

- All figures should be referenced in the main text and each figure should be referred to in order. Referring to figures in order will help with overall readability. For example Figure 3E is mentioned in the text before Figure 2.
- The figures have multiple subpanels eg Figure 2B1-8. Is this an acceptable format for journal?
- To improve readability, the paper would benefit from English grammar and spell checking

throughout. For example 2nd line in abstract reads "To performed systematic evaluation ...", should read "We performed" or "To perform a systematic evaluation"

- Line 88 mentions BRCA1/2 and EGFR actionable mutations and refers to Figure 1E – but neither of these genes are shown in this figure. Also what information was used to determine that the mutations were actionable?

Reviewer #3:

Remarks to the Author:

In this manuscript, the authors conducted a meta-study by integrating and re-analyzing the somatic mutations of 1616 ESCC genomes from the previous studies. The neoantigen prediction, mutational signature analysis and clinical-related assessments were performed base on the integrated dataset. The genomic features of ESCC including frequently mutated genes, mutational hotspots, neoantigen load, mutational signatures and pathways were re-analyzed in a larger cohort and related to the integrated clinical features, a combined test panel of eight genes for prognosis of ESCC were also proposed. The study is helpful to the field, by providing a more detail genomic landscape of ESCC which may facilitate the mechanism and clinical studies for ESCC.

However, the incomplete analyses of the variations across the genome (lacking the results of CNV, SV, virus integration etc.), have compromised the comprehensiveness of the study. And lacking of validation of the new finding by their own (or new) samples as well as by functional experiments have limited the novelty and importance of the study.

Major points:

1. As the dataset was collected from different published studies, the information of the sequencing type, mean coverage for each study should be provided and the normalization process of the data should be detailed.
2. Beside PMID29142225 and PMID3256150016, how the SNVs and indels were filtered and annotated in other datasets?
3. More details should be provided on the bioinformatic processing and mutation calling. The versions of the software used and the detail parameters should be provided.
4. For the analyses of TMB, to what extent could different rates of mutations calling affect the results? Does the difference of TMB in different stages or locations related to sequencing depths and coverages?
5. Line 81-84 (Figure 1F), for the mutated genes, ranking by the mutation frequency may not be enough to explore the contributions of the genes to tumorigenesis, other features such as gene length should also be taken into consideration. I recommend ranking the mutated genes by SMG instead.
6. The authors showed that the landscape presented prominent sites of mutational hotspots in chromosome 2, but no details were further discussed. Could the authors shed some light of the mechanism or functional impacts of this enrichment.
7. The authors combined eight genes to build the mutational score of ESCC, the significance of clinical outcome should be validated by independent cohort.

Minor points:

1. Please include the relevant colour keys in the figure panel for figure 1B, 4D, 5B and figure 7 left panel.
2. Please describe the results according to the order of the figures.
3. Line 245, how about the mutation rate of TNN in other cancer beside ESCC and stomach cancer. Does it specific in gastrointestinal tumors.
4. Line 296, I didn't find the gene with a HR<0 in figure 7.
5. Line 293, please explain what is the "mutational score" exactly mean, and how to calculate this value.

Reviewer #1, expert in immunogenomics and neoantigen prediction (Remarks to the Author):

Thank you for the opportunity to review the manuscript entitled, "Integrated cohort of esophageal squamous cell cancer revealed genomic features underlying clinical characteristics", for publication in Nature Communications.

ESCC is a common, poor-prognosis histologic subtype of esophageal cancer. ESCC biology is highly heterogenous, with poorly defined clinical subtypes and a lack of prognostic biomarkers, despite extensive whole-exome and -genome studies including several large patient cohorts. Better understanding and classification of the genomic etiologies of ESCC could lead to new targeted therapies or improved clinical management.

In the submitted work, the authors combine and jointly re-analyze 19 previously published ESCC cohorts and report on genomic features, potential neoantigens, and correlates of prognosis. The manuscript is mostly clearly written on a technical level. Sufficient details are included to permit reproducibility.

Major criticisms related to how the data support the proposed title- and abstract-level conclusions are:

FIGURE 1A-B

The critical, central technical issue of the authors' work is the extent to which the 19 previously published studies can be jointly re-analyzed. The concern is confounding due to differences in subject clinicopathologic composition and genomic data generation, from sample collection to variant calling. In particular, a notorious issue is the large degree of variability between different variant calling strategies in terms of criteria needed to consider a variant "real." If both clinicopathologic characteristics and genomic data generation pipelines differ across cohorts, the possibility of systematic bias arises.

Figure 1A shows that the distribution of non-synonymous mutations differs across cohorts, in many cases by a margin that appears likely to be statistically significant. Why is this - differences in subject characteristics, differences in data generation, or both? The null hypothesis is that both contribute. The degree of comparability of the 19 studies needs to be directly quantified by the authors, for instance by multivariate modeling that includes study ID as a variable, to understand that the results of the authors' re-analysis are valid. The t-SNE analysis of 1B does not adequately address this issue.

Response

Great thanks for your constructive comment.

We made the following efforts to minimized the potential influence of in sequence strategy,

analysis methods and data sources of clinical information among datasets, which was detailed explained in Methods part.

1, If the raw sequence data were also available, we directly re-analysis the raw sequence data, ignoring their published results. In the processing of raw sequence data, we used identical analysis and filtering pipeline.

2, In preparation of mutational records without raw sequence data, the authenticity of obtained variant list were firstly checked by comparison of the provided reference base to the loci in genome of it used, and then converted to hg38 genome.

3, After combination of all mutational lists, we filter all mutational records by the BED file used by TCGA project, which only contained basic exome regions and totally summed to 35.8 Mb. This capture region was smaller than all of the available WES capture size of included datasets (see sTable1). This step could minimize the influence by target strategy, and also make the final dataset more comparable to other TCGA cancer cohort.

Following your advice, we performed multivariate liner regression of TMB (sFigure1A, partly presented below, the largest dataset of PMID32398863 was set as reference), which indicated that, apart from one small sample dataset (PMID30012096, n=9), the sources of genomic dataset did not significantly influence mutational load.

FIGURE 3A:

The authors perform predicted neoantigen analysis by determining if peptides derived from somatic mutations are likely to bind with high affinity to HLA, which is thought to be a prerequisite for immunogenicity.

The authors' neoantigen analysis is incorrect and must be revised or removed from the manuscript. Peptide binding to HLA depends on HLA type, which must be determined for every subject either from sequencing data, for which accurate tools exist, or via another assay. The authors did not determine HLA types for any subjects. The authors instead used the "most common type I HLA alleles in East Asia." Given the large degree of variability across humans in HLA type, the analysis presented is inaccurate.

Response

Thanks for your comment, we had revised this part by prediction using the HLA-type from raw sequencing data of 413 patients. But the author group thought this part of analysis might be fully explained in another article and hope to remove this part from current work.

FIGURE 6B:

The authors combine mutation status for the eight most frequently mutated genes into a boolean composite mutation score. The biologic rationale of this is not clear. Such a mutation score is likely to be highly correlated with, and thus a surrogate marker for, overall tumor mutation burden. The prognostic significance of TMB is well-studied.

Response

Thanks for your professional comment. Under your advice, we tested the association between the mutational score and total mutational load. Not unexpectedly, the mutational score was positive associated with total TMB (Figure5D).

However, the TMB value did not have prognostic significance in ESCC-META cohort(sFigure1D).

Reviewer #2, expert in cancer genomics and mutational signatures (Remarks to the Author):

The manuscript titled: “Integrated cohort of esophageal squamous cell cancer revealed genomic features underlying clinical characteristics” describes a meta-analysis of the somatic SNV and indel mutations in a cohort of 1616 ESCC samples from 19 studies. Through this meta-analysis the authors aim to determine prognostic mutation features.

There are number of major concerns about this paper which affect the robustness, impact and overall conclusions:

- The novelty of the study is not clear. What added advantage has this meta-analysis shown over previous studies of ESCC?

Response

Thanks for your comment. We thought the novelties of the revised article could be summarized in the following three parts.

1, Compared with previous single cohort or integrated studies, the ESCC-META cohort had significant advantages in quality control, sample size and its open source and updating characters. Influenced by the high heterogeneity of the genomics of ESCC, the previous studies of mutational signature reported varied results. In the mutational signature, we used 1084 WGS data of ESCC as discovery set, which identified 11 signatures, and further accessed their relevance with clinical variables in 1930 patients. We thought our result provided the most credible evidence in this field by far.

2, We identified many significant mutated genes and hotspots (such as *TGFBR2* and *IRF2BPL*) that had not been reported before. We noticed the different prognostic impacts of some mutations in early or late stage and identified many novel prognostic mutated genes (such as *ZNF721* and *LAMA3*). For the first time, we discovered the different mutational profile between ESCC from upper thoracic part and lower thoracic part.

3, We proposed the concept of prognostic mutational score, and using our own data as testing set, we proved the efficacy of the eight genes mutational score in ESCC.

- The analysis of frequently mutated genes is anecdotal. Approaches to look at significantly mutated genes, as opposed to the frequency of mutations would improve this section. Frequency of mutations within genes can be affected by things such as gene size

or hotspots for example the TTN gene is very large and may not be significant. The authors should consider running methods such as Intogen and MutSigCV (there are others also) to determine what are the significantly mutated genes, otherwise the findings are not robust.

Response

Thanks for your comment. We have added the results of MutSig2CV in revised manuscript (Figure1D, Figure3A and sTable7).

In the ESCC-META cohort of 1930 patients, totally 1888 genes mutated in more than 1% patients, among whom 761 genes (40.3%) could be identified as significantly mutated genes by MutSig2CV (q value <0.05). Although we had excluded patients without available synonymous mutational records to avoid the false positive results, the q value is very small in the majority of top mutated genes, which we thought could be contributed to the large size of sample we employed in analysis.

We still presented the mutational profile by the order of frequency, but added a panel in right side to indicate their log scaled q value from MutSig2CV.

- The genomic distribution of mutational hotspots section looks for recurrent genomic positions. For recurrent positions many were present in just 2 samples. It would be helpful to know how many of these recurrent positions were only present in 2 samples and how confident are the authors that these are present by chance or passenger events as

opposed to driving events. The approach to look at recurrent loci can also be improved by looking at significantly mutated loci (see previous comment).

Response

Thanks for your comment. In the updated ESCC-META dataset, we totally identified 188,847 non-silent SNVs, among which 179,531 were unique, and only 6917 of them (3.88%) could be detected in two or more patients.

Due to the complexity of genome, it was hard to accurately calculate the expected rate of recurrent only by chance. Here we provide a conceptual estimation using binomial distribution. There are 6,034 SNVs in just 2 samples and we assume they are totally by chance. The following table presented the calculated expected rate by chance (equally distributed sampling m times from N possible mutational sites with replacement) and actual observed rate.

$$p_k = \binom{m}{k} \frac{(N-1)^{m-k}}{N^{m-1}}$$

$$\frac{p_k}{p_{k+1}} = \frac{n_k}{n_{k+1}} = \frac{(N-1) * (K+1)}{m-k}$$

Times of recurrence	Observed Number of SNVs	Expected Number by chance
1	172,560	172,560
2	6,034	6,034
3	618	105.5

4	119	1.84
5	67	0.026
6	34	0.00036
7	21	3.604862e-06
8	17	<0.000001
9	7	<0.000001
10	12	<0.000001
>10	42	<0.000001

In the above table, we could see significantly higher chance of observed recurrent SNVs than the expected number in totally random sampling.

We also had calculated the mutational density in all CDS by sliding window of 200bp, but we did not present this analysis in our manuscript because this process is not so rigor and might be misleading.

On the other side, we though the results from MutSig2CV were more credible in identifying significant mutations.

In the revised manuscript, we paid attention to the most significant and hotspots, such as the recurrent frame shift deletion in *TGFBR2* (c.374delA) and *IRF2BPL* (c.224_305del), each of whom could identified from different patients and from different studies, thus could hardly be explained by technical error.

- The analysis and results presented lack detail making them hard to interpret. As an

example, for the TSNE in Figure 1B there are no axis labels, it is not clear what numeric data is being used in the plot (is it absence presence of mutations in specific genes?) and the legend does not contain sufficient detail.

Response

Thanks for your comment. We had made many improvements to explain the details in revised manuscript. The t-SNE analysis was performed by the mutation matrix of all integrated genomes of top 1000 genes.

- The findings from this paper hinge on there being significant mutations and survival differences between groups. However, often the statistical test used is not defined in the text or in the figure legend, the p-values reported are sometimes low, the test may be confounded by other factors such as grade or age, and the size of the groups in some of the analyses are unclear - for example in the survival plots in Figure 2D2.

Response

Thanks for your comment. We had provided all statistical details in revised manuscript. The labelled p value in survival curves were obtained by log rank test, and the significance of comparison was also validated in multivariable Cox analysis.

- The mutational signature work needs more detail. Does the meta-analysis find new signatures related to ESCC? Were all mutations used in the analysis (recommended) or

just a subset as indicated from line 382-385. In Figure 4D the authors aim to show the correlation of signatures with age, however this would be clearer as a scatter plot of signature contribution vs age, since it is hard to draw conclusions from the proportion plots.

Response

Thanks for your comment. With the help of newly published WGS cohort of ESCC, we had performed new mutational signature analysis and identified 11 mutational signatures from 1084 WGS samples, and predicted their contributions in ESCC-META cohort.

The age-related signature(sig2) was presented in Figure2H as a scatter plot, which indicated strong positive relationship.

- For the TMB and neoantigen section in Figure 3E and F, how were the groups of “low” and “high” defined? Why was a TMB of 5 selected - is this clinically relevant? The p-value for neoantigen is very low (Figure 3F) at 0.034, was this also correct for other clinical

factors such as age, or cohort?

Response

Thanks for your comment. In the revised manuscript, the TMB were grouped to high or low by the median value for survival comparison, which indicated no significant difference.

The median number of nonsynonymous mutations in the ESCC-META was 81 (52 of 25th percentiles and 117 of 75th percentiles), which corresponded to the estimated median tumor mutation burden (TMB) of 2.26 mutation per Mb. About 13.7% tumors presented TMB lower than 1, while 8.6% and 1.8% patients reached or above 5 and 10 of TMB, respectively.

As the first reviewer pointed, the previous neoantigen analysis was not solid enough, and the updated results of this part might be fully explained in another article. The author group hope to remove this part of analysis from current work.

- The prognostic signature based on mutations in 8 genes, could this also be cofounded by other clinical features? How clinically useful would this signature be? What number of samples harbor mutations of these genes? It would be good to validate this 8 gene signature in another cohort, since it is a key finding in the paper.

Response

Thanks for your comment. In the revised manuscript, the mutational score was well illustrated and tested among different dataset by multivariable Cox analysis.

We also applied the our own ECRT datasets as independent testing set, which could provide evidence of higher quality because its patients were all participants of a large phase III ESCC clinical trial (ChiCTR-IPR-15007172) with prospective and homogeneous enrollment, treatment and follow-up. The mutational score performed well as an effective predictor for bad prognosis in testing set (HR of one mutation = 2.21, HR of two all more mutation= 8.02, log rank p value=0.0023).

Minor comments

- All figures should be referenced in the main text and each figure should be referred to in order. Referring to figures in order will help with overall readability. For example Figure 3E is mentioned in the text before Figure 2.
- The figures have multiple subpanels eg Figure 2B1-8. Is this an acceptable format for journal?
- To improve readability, the paper would benefit from English grammar and spell checking throughout. For example 2nd line in abstract reads “To performed systematic evaluation ...”, should read “We performed” or “To perform a systematic evaluation”
- Line 88 mentions BRCA1/2 and EGFR actionable mutations and refers to Figure 1E – but neither of these genes are shown in this figure. Also what information was used to determine that the mutations were actionable?

Response

Thanks for your comment. These details had been corrected in revised manuscript.

The BRCA1/2 and EGFR actionable mutations could be seen in sFigure3A of the revised manuscript.

Reviewer #3, expert in ESCC genomics (Remarks to the Author):

In this manuscript, the authors conducted a meta-study by integrating and re-analyzing the somatic mutations of 1616 ESCC genomes from the previous studies. The neoantigen prediction, mutational signature analysis and clinical-related assessments were performed base on the integrated dataset. The genomic features of ESCC including frequently mutated genes, mutational hotspots, neoantigen load, mutational signatures and pathways were re-analyzed in a larger cohort and related to the integrated clinical features, a combined test panel of eight genes for prognosis of ESCC were also proposed. The study is helpful to the field, by providing a more detail genomic landscape of ESCC which may facilitate the mechanism and clinical studies for ESCC.

However, the incomplete analyses of the variations across the genome (lacking the results of CNV, SV, virus integration etc.), have compromised the comprehensiveness of the study. And lacking of validation of the new finding by their own (or new) samples as

well as by functional experiments have limited the novelty and importance of the study.

Major points:

1. As the dataset was collected from different published studies, the information of the sequencing type, mean coverage for each study should be provided and the normalization process of the data should be detailed.

2. Beside PMID29142225 and PMID3256150016, how the SNVs and indels were filtered and annotated in other datasets?

3. More details should be provided on the bioinformatic processing and mutation calling.

The versions of the software used and the detail parameters should be provided.

Response

Thanks for your comment. The above 3 questions had been detailed explained in revised manuscript.

We established a set of pipelines for data selection and process to build integrated genomic cohort, and in current ESCC-META dataset, 413 patients from 15 datasets (including our own sequence data) were re-analysis from raw reads data, and the rest somatic mutational records of 1517 patients from 18 datasets were prepared from published mutational list (sTable1 and sTable2).

For raw sequence data

If the reads data were sra format, it was firstly converted to fastq file by SRA-Tools. All fastq file were performed quality control by fastp(v0.23) by default parameters, and all paired runs or sequence lanes from the same sample were combined before mapping.

The mapping was performed by BWA (v0.7.1) to hg38.p13 genome. The duplicated reads were further removed and the rest data were applied base quality score recalibration by GATK (v4.1). The obtained bam files were firstly examined by bam-matcher to exam the pair relationship between tumor and normal samples labelled to one patient. Several mismatched data were removed (see Supplementary Materials).

The single nucleotide variants (SNVs) and insertion or deletion mutations (INDELs) were called by Mutect2. The filter criteria included the following three situations.

1, for WES sequence of one tumor sample with normal control, the coverage should be at least 30 in tumor sample and at least 20 in normal sample, at least three alternative reads in tumor sample to support the variant call, and mutation frequency at least 0.05.

2, for WGS sequence of one tumor sample with normal control, the coverage should be at least 20 in tumor sample and at least 20 in normal sample, the rest is the same as above.

3, for WES sequence of multiple tumor samples with one normal control from the same patient the included variant should be detected in at least one tumor sample meeting the above type 2 criteria, additionally the variant base should be identified (at least two

alternative support reads) in another tumor sample.

For preparation of mutational records

For mutational records from reported list (such as in MAF format) or database without raw data, they were firstly prepared in the following two steps.

1, the authenticity of obtained variant list were firstly checked by comparison of the provided reference base to the loci in genome of it used. For example, if the raw mutational record is chr19:63554635, G>T in hg18, the true base of chr19:63554635 in hg18 should be G, if not, this record was suspicious and must be re-examined.

2, the verified records from each dataset were prepared to unified vcf format and liftover to hg38 by CrossMap(v0.2.6). This step will also remove a few records because of failure to convert. The converted mutational lists were re-check with hg38 as the first step.

Integration and Annotation

The patients all were renamed by pasting their source dataset before their original IDs.

The duplicated samples were carefully identified by checking their source information and by pairwise mutational profile comparisons, only the one original sample remained in the final inclusion.

The results from our analysis pipeline (passed VCFs files) and converted mutational list were then combined and annotated by ANNOVAR (December 2019 version). This overall combined dataset was used in mutational signature analysis, while for the rest analysis,

we used regional filtered dataset. We filter all mutational records by target region BED file (converted to hg38) used by TCGA. This BED file only contained basic exome regions and totally summed to 35.8 Mb, which was smaller than all of the included WES capture regions.

4. For the analyses of TMB, to what extent could different rates of mutations calling affect the results? Does the difference of TMB in different stages or locations related to sequencing depths and coverages?

Response

Thanks for your comment. We performed multivariate linear regression of TMB (sFigure1A, partly presented below), which indicated that, apart from one small sample dataset (PMID30012096, n=9), the sources of genomic dataset did not significant influence TMB. There was also no difference between sequence type (WES or WGS), and thus we thought in the quality-controlled final dataset, the sequencing depths and coverages were not confounding factors to TMB.

5. Line 81-84 (Figure 1F), for the mutated genes, ranking by the mutation frequency may not be enough to explore the contributions of the genes to tumorigenesis, other features such as gene length should also be taken into consideration. I recommend ranking the mutated genes by SMG instead.

Response

Thanks for your comment. We have added the results of MutSig2CV in revised manuscript (Figure1D, Figure3A and sTable7).

In the ESCC-META cohort of 1930 patients, totally 1888 genes mutated in more than 1% patients, among whom 761 genes (40.3%) could be identified as significantly mutated genes by MutSig2CV (q value <0.05). Although we had excluded patients without

available synonymous mutational records to avoid the false positive results, the q value is very small in the majority of top mutated genes, which we thought could be contributed to the large size of sample we employed in analysis.

We still presented the mutational profile by the order of frequency, but added a panel in right side to indicate their log scaled q value from MutSig2CV.

6. The authors showed that the landscape presented prominent sites of mutational hotspots in chromosome 2, but no details were further discussed. Could the authors shed some light of the mechanism or functional impacts of this enrichment.

Response

Thanks for your comment. The density of non-silent SNVs in a genomic region was influenced by the density of CDS and the functions of important genes.

Chromosome	Total non-silent SNVs	length	N per Mb
chr1	18962	248,956,422	76
chr2	14761	242,193,529	60
chr3	11307	198,295,559	57
chr4	7521	190,214,555	39
chr5	9080	181,538,259	50
chr6	9657	170,805,979	56
chr7	9613	159,345,973	60

chr8	7873	145,138,636	54
chr9	6497	138,394,717	47
chr10	6912	133,797,422	52

The highly mutated genes in chromosome 2 included REG3A, LRP1B, XIRP2, NFE2L2, TTN and FSIP2, among which REG3A and NFE2L2 were more important.

The mutational hotspots of REG3A(2p12) and NFE2L2(2q31.2) should be gain-of function, which was consistent with the oncogenic functions of the two gene.

However, we did not find previous evidence of copy number alteration of this chromosome in ESCC. The discussion of the chromosome in other solid tumors was also rare. We are sorry that we could not provide more explanations of this enrichment.

7. The authors combined eight genes to build the mutational score of ESCC, the significance of clinical outcome should be validated by independent cohort.

Response

Thanks for your comment. We applied the our own ECRT datasets as independent testing set, which could provide evidence of higher quality because its patients were all participants of a large phase III ESCC clinical trial (ChiCTR-IPR-15007172) with prospective and homogeneous enrollment, treatment and follow-up. The mutational score performed well as an effective predictor for bad prognosis in testing set (HR of one mutation = 2.21, HR of two all more mutation= 8.02, log rank p value=0.0023).

Minor points:

1. Please include the relevant colour keys in the figure panel for figure 1B, 4D, 5B and figure 7 left panel.
2. Please describe the results according to the order of the figures.
3. Line 245, how about the mutation rate of TNN in other cancer beside ESCC and stomach cancer. Does it specific in gastrointestinal tumors.
4. Line 296, I didn't find the gene with a $HR < 0$ in figure 7.
5. Line 293, please explain what is the "mutational score" exactly mean, and how to calculate this value.

Response

Thanks for your comment. These issues were all corrected in revised manuscript.

Reviewers' Comments:

Reviewer #1:

Remarks to the Author:

Thank you for the opportunity to review the revised manuscript entitled, "Integrated cohort of esophageal squamous cell cancer revealed genomic features underlying clinical characteristics", for publication in Nature Communications.

Thank you to the authors for their additional effort.

The authors have adequately addressed all of my concerns.

Reviewer #2:

Remarks to the Author:

Thanks to authors for taking the time to address my comments. However, I still have several concerns regarding the paper.

1. The addition of MutSig2CV analysis to identify significantly mutated genes adds weight to the description of mutated genes. However, even though MutSig2CV has been run the authors still refer to the genes with the highest number of mutations and select these genes to plot in Figure 1D. As stated previously this approach is not ideal. A large gene is more likely to have more mutations and therefore the total number of mutations is misleading. I would suggest focusing the early results on those in Figure 1 on genes that are significantly mutated in the cohort. I also have a number of comments with the MutSigCV analysis:

- There are >800 hundred genes with a Q-value of <0.01 (shown in excel file tab "sheet 10"), this seems like a surprisingly large number of genes and the authors also note this in the rebuttal. Can the authors comment more on why so many genes are significant? Can you review and provide more details in the methods section of how MutSig2CV was run? And how to interpret the tab labelled "sheet 10" in the excel file of Supp tables. Could the authors other tools to identify significantly mutated genes and focus on genes identified by multiple tools?

- The MutSig data has been added to Figure 1D and "sheet10" in the excel file. But this data or result is not referred to in the text. Please refer to Figure 1D in the text.

2. Not all Figures are mentioned in the main text, for example Figure 1C. Not all figures are mentioned in order in the text. Please refer to all Figures in the main text at the appropriate times when the results are described. Please check all figures are referred to in order in the main text to help with flow of the paper.

3. The Figure legends within the paper would still benefit from more detail to clarify what they are showing. For example:

a) In Fig 1D what is the "Multi-Hit" group? In Figure 1D most samples appear to have genes mutated with a Multi-Hit (indicated by green in oncoplot), however in the plot on the right the mutational frequency shows many more missense mutations (blue). Is this correct? Is it expected that many genes will be mutated by multi-hit in this tumour type?

b) Also in Figure 1D and 3A it states the "top genes" were selected, are the top genes the most frequently?

c) Figure 3E separates patients by 'late' and 'early', assume this is late and early stage of disease?

4. In the mutational signature analysis sig4 which is similar to COSMIC3 was detected in about 10% of samples (Figure 2E). The authors rule out an association with somatic mutations in BRCA1/2 (Figure S2C). However, the presence of COSMIC3 is interesting as it may indicate HR deficiency which could have treatment implications. Running an approach such as HRDetect (which takes into other

signatures to identify HR deficiency) would be useful to determine if the tumours are HR deficient.

5. Cluster 1 linked to sig1 (Figure 2E) shows worse survival. In addition to a link with drinking and smoking (as mentioned in results) there is also an association with sex (Figure 2G). Is sex associated with survival? , so could sex be what is causing the poor survival in this group.

6. The TMB was calculated, using non-synonymous SNVs in a 35.8Mb capture region. Was the 35.8Mb capture region part of the regions targeted by all capture platforms? The dataset PMID30012096 appears different to the others in terms of mutation burden (Supp Figure S1A). Can the authors add some discussion as to why this may be the case?

7. The pathway work in Figure 3 is interesting but could be improved. The pathway analysis on Line 174 says "Although most genes presented low mutational frequency in ESCC, their related functions were enriched in several major oncogenic pathways". The current analysis does not support this statement as it does not show that the pathways were enriched in this data, only that they are frequently mutated. This could be caused by the pathways having a large number of genes, or the genes in these pathways being large and more likely mutated. Additionally "Around 14.7% patients carried at least targetable mutations, such as BRCA1/2 (5%) and EGFR (2%) (sFigure3A)". How were mutations classified as "targetable"?

8. Figure 3D shows that drinkers and smokers have more EP300 changes and this more hotspot changes. Could this just be due to a higher mutation load in the patients who are drinkers and smokers? Line 213 says: "and associated with worse prognosis compared with mutations in other EP300 regions or wide types (Figure3E)", note the EP300 mutations were only linked to prognosis in late stage disease.

9. Some of the results have changed in the revised version. Previously the authors reported on 1616 samples, and now there are 1930 samples, also "The median number of nonsynonymous mutations in the ESCC-META was 74 (Figure 1A), which corresponded to the estimated median tumor mutation burden (TMB) of 2.07 mutation per Mb". The revised version says: "The median number of nonsynonymous mutations in the ESCC-META was 81 (52 of 25th percentiles and 117 of 75th percentiles), which corresponded to the estimated median tumor mutation burden (TMB) of 2.26 mutation per Mb". What is the reason for these changes? Is it because of the revised number of samples and studies used?

10. Another change is the opening sentence of the results which says "We established a set of pipelines for data selection and process to build integrated 55 genomic cohort (see Methods for details)," If establishment of these pipelines is a key result then the pipelines and code should be made available. A key feature of this manuscript is the collation of the somatic mutations for the 1930 cases, it would be very helpful if this was made available to the readers.

11. The description of the 8 gene signature is important for the novelty of the paper and the inclusion of the validation data for the 8 gene signature has improved the paper. However, due to the importance of the signature in the paper, it would be really good if the authors could address my original comment and add to the discussion how this would be implemented clinically and whether it is better than any clinical or other markers of prognosis that may be used.

Other comments

1. Line 268 says "The mutational rate of CSMD3, PCLO, NFE2L2 and FLG were significantly higher in ESCC of upper thoracic part, while the mutation of NOTCH1, MUC4 were more common in lower thoracic part (sTable7)", please add a pvalue between these groups to show significance

2. My previous comment asked how the groups of "low" and "high" TMB were defined. Although this was answered in the rebuttal, please add this to the relevant figure legends, e.g. in Figure S1D.

3. Please review the manuscript for readability and grammar throughout the manuscript. For example:
- a) Line 59 says "were re-analysis from raw reads data", should read: were re-analysed from raw reads data
 - b) Line 101 says "Forest plot of the of the", remove: of the, which is listed twice
 - c) Line 214 says "wide types", should be: wild types
 - d) Line 285 says "sFigur4C" and line 286 says "Figur4E", replace with: Figure
 - e) Line 338 says "ESCC clinical trial (xxxxxxx)" – please correct this text to specify what is meant by xxxxxxx
 - f) Line 436 says "phase III clinical trial of xxxxxxx" – please correct this text to specify what is meant by xxxxxxx
 - g) Line 507 says "The MutSig2 were used", should read: The MutSig2 approach was used
 - h) Line 508 says "were exclude", should read: were excluded
 - i) Line 509 says "non-silence mutations", please change to: non-silent
 - j) Line 516 says "to decomposes major mutational signature", should read: to decompose major mutational signatures
 - k) Line 517 says "The optimal number of separations was selected both considering the most decline in cophenetic correlations and the afford size of residual sum of squares" – the meaning of this sentence is not clear, please revise
 - l) Line 523 says "The COSMIC Mutational Signatures database (v2) were", should read: The COSMIC Mutational Signatures database (v2) was
 - m) Line 557 says "The meta-analysis of single proportions in random effect model was 558 also employed to systematically assess the pooled mutational rate of single gene", – the meaning of this sentence is not clear, please revise
 - n) Please review and edit the rest of the manuscript, in particular the methods section.

Reviewer #3:

Remarks to the Author:

In the revised version, the authors have addressed all the questions I raised. I have no more question.

Reviewer #2 (Remarks to the Author):

Thanks to authors for taking the time to address my comments. However, I still have several concerns regarding the paper.

We are truly grateful for your help to improve the quality of the manuscript. We benefited greatly from these suggestions. Your professional work means a lot to our team.

In this version of revision, we mainly achieved four improvements.

1) According to your suggestion, we used combined tools to detect the most significantly mutated genes and identified 22 important genes. This part was added to the manuscript and explained in the following response.

2) Thanks to your comment, we performed systematic evaluations on the heterogeneity of WES capture platforms. The estimation indicated no more than 1% non-synonymous SNVs in test set would be dropped in WES capture platforms compared with WGS sequence. There were several genes with potential research values out of the capture range of some platforms. We listed these genes in sFigure2 and sTable4 warned that their mutational rates might be underestimated.

The genes that located in incomplete covered regions by one or some capture platforms were excluded from building mutational score to avoid potential artifacts in survival analysis. This improvement altered the panel of mutational score because the previously included gene of *MUC4* was excluded.

3) We remove the estimation of tumor mutational burden (TMB) in the revised manuscript, which was greatly influenced by its denominator of total capture length and would be misleading in direct comparisons between different platforms. We encouraged readers to estimate the TMB value based on their own target region using our provided function.

4) Thanks for your suggestions. We had greatly improved readability of the total manuscript. In particular, we had presented more detailed and clearer explanations in Methods sections.

1. The addition of MutSig2CV analysis to identify significantly mutated genes adds weight to the description of mutated genes. However, even though MutSig2CV has been run the authors still refer to the genes with the highest number of mutations and select these genes to plot in Figure 1D. As stated previously this approach is not ideal. A large gene is more likely to have more mutations and therefore the total number of mutations is misleading. I would suggest focusing the early results on those in Figure 1 on genes that are significantly mutated in the cohort. I also have a number of comments with the MutSigCV analysis:

- There are >800 hundred genes with a Q-value of <0.01 (shown in excel file tab "sheet 10"), this seems like a surprisingly large number of genes and the authors also note this in the rebuttal. Can the authors comment more on why so many genes are significant? Can you review and provide more details in the methods section of how MutSig2CV was run? And how to interpret the tab labelled "sheet 10" in the excel file of Supp tables. Could the authors other tools to identify significantly mutated genes and focus on genes identified by multiple tools?
- The MutSig data has been added to Figure 1D and "sheet10" in the excel file. But this data or result is not referred to in the text. Please refer to Figure 1D in the text.

Response: Really thanks for your comments, we have revised our results according to your suggestions.

We have moved the frequency ranked oncoplot of Figure 1D to sFigure1, and we added the following sentence in result part:

"The most frequently mutated genes (sFigure1B) would not necessarily suggest their important contributions to ESCC tumorigenesis, because many genes with high mutational frequencies might owe to their great gene length, such as TTN and MUC16. However, these genes could help us to assess the homogeneity among datasets."

1, Combined tools to identify significantly mutated genes.

In the revised manuscript, we used the following four approaches to identify significantly mutated genes. The details were explained in the Methods part "Identification of significant mutated genes"

• Identification of significant mutated genes⁴¹

Genomes from three studies (PMID22877736, n=12; PMID32929369, n=14; PMID28608921, n=41) were excluded in this part of analysis because these records only contained non-silent mutations and without available synonymous mutations, which might increase the false positive rate in integrated analysis. The remaining 1863 ESCC genomes were included. Besides mutational frequency, we used the following four approaches to identify most important genes. The 2,3,4 approaches need specified transcript for each gene, and we selected the most common transcript based on the SNVs annotation results.⁴¹

↵

1, We applied MutSigCV to calculate the Q value. MutSigCV was designed to identify genes that were mutated more often than expected by chance given background mutation processes⁵⁶. We prepared the MAF file to comply with the requirements, and the other input files of coverage table, covariates table and mutation type dictionary file were downloaded from CGA (https://software.broadinstitute.org/cancer/cga/mutsig_run) under its directions.⁴¹

↵

2, We applied oncdriverCLUST to calculate the Cluster score. The oncdriverCLUST was designed to find driver genes with enriched mutational hot-spots⁵⁸. We used this tool with default parameters.⁴¹

↵

3, We calculated the ratio of non-synonymous mutation to synonymous substitution (dN/dS) in the coding region (CDS) for each gene. The high dN/dS values suggested positive selection in cancer evolution⁵⁷.⁴¹

↵

4, We calculated the coding length adjusted mutational frequency for each gene. We defined this value as mutational density in this article.⁴¹

$$\text{mutational density} = \frac{\text{N of synonymous mutations/N of total patients}}{\text{length of CDS(Mb)}}$$

↵

We applied the following five criteria to obtain the most significant genes: 1, mutational frequency $\geq 2\%$; 2, MutSigCV q value ≤ 0.01 ; 3, OncdriverCLUST clusterScores ≥ 0.2 ; 4, Mutational density ≥ 50 ; 5, dN/dS ≥ 5 .⁴¹

Totally 22 genes were identified as significant genes. The new results were presented in Figure4A, Figure4B (the following figures) and sTable9.

2, Why so many genes are significant in MutSigCV?

We had carefully examined the process of MutSigCV analysis and confirmed our results. We thought there were two reasons for so many significant genes.

1) The MutSigCV was designed to identify genes that were mutated more often than expected by chance given background mutation processes. In a large genomic group from same disease, the number of identified significant mutated genes would increase because with increased statistical efficiency by large sample size, the relative weak effects (mutations of low frequency) would be significant.

We used 1863 ESCC genomes in MutSigCV analysis, and this sample size should identify more

significant genes under its default parameters.

2) In ESCC-META dataset, we totally detected non silent SNVs in 18097 genes, among which only 1888 genes (10.4%) mutated in more than 1% patients. In the results of MutSigCV analysis, we could see that although more than 800 genes presented Q-value of <0.01 , they were still a small proportion of total altered genes (the following table and figure).

Mutational frequency	Number of genes	Number Q-value <0.01
$\geq 50\%$	1	1(100%)
$\geq 10\%$	9	8(89%)
$\geq 5\%$	50	40(80%)
$\geq 2\%$	434	236(54%)
$\geq 1\%$	1888	565(29.9%)
$>0\%$	18097	913(5%)

If we filtered the input genomes before MutSigCV analysis, such as only selecting mutations in more than 1% frequency genes, we would get less significant genes by Q-value (or false discovery rate, which controls the percentage to be significant), but it is not the routine practice and might induce other biases.

3, Detailed explanation of the process in MutSigCV analysis.

1) the samples in PMID22877736, PMID32929369 and PMID28608921 were excluded from this analysis because they only included non-silent mutational records. The remaining 1863 ESCC genomes all contained both non-synonymous mutations and synonymous mutations.

2) Because the recommended covariates files in MutSigCV analysis used some old gene names, we used the prepareMutSig function in maftools package to convert the original Hugo_Symbols to that compatible with MutSigCV. The gene names in MutSigCV result were reversely transformed to standard gene names.

ref <https://rdrr.io/bioc/maftools/man/prepareMutSig.html>

3) The MAF file were converted to hg19 using CrossMap and prepared for MutSigCV process.

4) Except the prepared MAF file, the other three input files were download from CGA (https://software.broadinstitute.org/cancer/cga/mutsig_run) under its directions.

coverage table file

http://www.broadinstitute.org/cancer/cga/sites/default/files/data/tools/mutsig/reference_files/exome_full192.coverage.zip

covariates table file

http://www.broadinstitute.org/cancer/cga/sites/default/files/data/tools/mutsig/reference_files/mutation_type_dictionary_file.txt

mutation type dictionary

http://www.broadinstitute.org/cancer/cga/sites/default/files/data/tools/mutsig/reference_files/gene.covariates.txt

The MutSigCV analysis on hg19 was then performed.

2. Not all Figures are mentioned in the main text, for example Figure 1C. Not all figures are mentioned in order in the text. Please refer to all Figures in the main text at the appropriate times when the results are described. Please check all figures are referred to in order in the main text to help with flow of the paper.

Response: Thanks for your kindly comment. We are sorry for this carelessness. We had corrected these mistakes in this revised submission.

3. The Figure legends within the paper would still benefit from more detail to clarify what they are showing. For example:

a) In Fig 1D what is the "Mulit-Hit" group? In Figure 1D most samples appear to have genes mutated with a Multi-Hit (indicated by green in oncoplot), however in the plot on the right the mutational frequency shows many more missense mutations (blue). Is this correct? Is it expected that many genes will be mutated by multi-hit in this tumour type?

b) Also in Figure 1D and 3A it states the "top genes" were selected, are the top genes the most frequently?

c) Figure 3E separates patients by 'late' and 'early', assume this is late and early stage of disease?

Response: Thanks for your kindly comment. We had improved the explanations in this revised submission.

a) The Multi-Hit represents two or more non-silent mutational sites of the specified gene in one patient. Such as the gene *TP53*, total 1509 patients (78%) had at least one *TP53* mutation, among whom 233 patients had two or more mutational sites (the following table).

N of TP53 mutations	Number of patients
--------------------

0	421
1	1276
2	227
3	5
4	1

In the revised manuscript, we had modified the colors of all oncoplots. The “Multi-Hit” was colored in black for better distinction.

b) Yes, the top genes are the most frequently

c) Yes, the 'late' and 'early' referred to the stage of disease.

4. In the mutational signature analysis sig4 which is similar to COSMIC3 was detected in about 10% of samples (Figure 2E). The authors rule out an association with somatic mutations in BRCA1/2 (Figure S2C). However, the presence of COSMIC3 is interesting as it may indicate HR deficiency which could have treatment implications. Running an approach such as HRDetect (which takes into other signatures to identify HR deficiency) would be useful to determine if the tumours are HR deficient.

Response: Thanks for the constructive comment.

We have performed some further analysis, but the results seemed not solid enough to draw a reliable conclusion. The following presented a detailed explanation.

1, SBS3 and ID6 associated with HRD

The sig4 was similar to COSMIC3 (or SBS3, similarity=0.90), COSMIC5 (similarity=0.80), COSMIC16(similarity=0.72) and COSMIC8 (similarity=0.72), among which the COSMIC

3/16/8 were related to DNA repair. The SBS3 presented patterns of evenly distributed base substitutions types, which was also featured in sig4 (as the following picture).

The SBS3 was associated with HRD. Another characteristic signature of HRD was ID6 (Alexandrov et al. 2020 Nature), which shown elevated numbers of large (longer than 3bp) insertions and deletions (as the following figure).

2, The sig4 was also related to ID6.

We further extracted the ID83 matrix of subclass categorization of ID (AlexandrovLab/SigProfilerMatrixGenerator) and calculated the ID6 activity (AlexandrovLab/SigProfilerExtractor) in the WGS ESCC samples. The activities of sig4 and ID6 were positively correlated (as the following picture).

3, The analysis of small insertions and deletions was less reliable in integrated dataset.

However, the batch effects among studies in terms of small insertions and deletions (ID) could not be ignored.

The following figures explained the difference in ID among the four datasets (all of them were WGS results). While there was no significant batch effect in SBS96 matrix (Figure2 in manuscript, right part of the following figure), Both the ID83 count matrix and proportion matrix shown significant batch effects among studies in tSNE (left part of the following figure).

We had noticed significant heterogeneity in small insertions and deletions (ID), copy number variants (CNV) and structures variants (SV) in our analysis. Due to the lack of effective approaches to suppress these batch effects, we excluded this analysis from our works. We thought that, compared with the accuracy and reproducibility in detection point mutations, the detection of large-scale variants might be less robust in the current NGS platform, which was more sensitive to the capture platform, sequence mode and data quality, the coverage of the region, and the filter parameters.

4, We could not run approaches such as HRDetect

The HRDetect pipeline (Nik-Zainal-Group/signature.tools.lib) needs six features to estimate HRD status:

- 1) proportion of deletions at microhomology (del.mh.prop);
- 2) number of mutations of substitution signature 3 (SNV3);
- 3) number of mutations of rearrangement signature 3 (SV3);
- 4) number of mutations of rearrangement signature 5 (SV5);
- 5) HRD LOH index (hrd);
- 6) number of mutations of substitution signature 8 (SNV8).

The 1) and 5) were not available for most samples, the 3) and 4) features were not so reliable in the integrated dataset (as explained above).

We had tried several other exploratory analyses, but could not obtain convincing results due to limited reliable supporting data. Essentially, the genomic HRD assessment more relied on the detection of large-scale genomic variants, such as LOH, genomic scars and large insertions and deletions. Our current work focus on the ESCC point mutations, thus could not provide a direct answer to the question.

5, Discussion

The germline BRCA1/2 variants could identify in around 3% of ESCC cases, and the somatic variants were around 5%. The HRD might contribute to mutational load in some ESCC cells thus we could detect the SBS3 and ID6, which was also identified in other ESCC studies (doi: 10.1038/s41588-021-00928-6. Nature Genetics, 2021, the following figure).

However, the genomic heterogeneity and technical reasons limited our further association discoveries. The combinations of SBS3 and TMB might be more accurate to predict the HRD status, which need further evidence to verify.

5. Cluster 1 linked to sig1 (Figure 2E) shows worse survival. In addition to a link with drinking and smoking (as mentioned in results) there is also an association with sex (Figure 2G). Is sex associated with survival? , so could sex be what is causing the poor survival in this group.

Response: Thanks for your comment. Sex could influence survival (not significant), but could not explain the worse prognosis of patients with sig1. We performed a multivariable cox regression in all patients (the following plot, sFigure3E in the revised manuscript). The HR values were presented in the following figure, which suggested Cluster1(patients with predominant sig1 mutations) as an independent prognostic factor (the line in red box, HR=1.37, p=0.016) regardless of age, sex, smoking, drinking, and tumor stage.

E

6. The TMB was calculated, using non-synonymous SNVs in a 35.8Mb capture region. Was the 35.8Mb capture region part of the regions targeted by all capture platforms? The dataset PMID30012096 appears different to the others in terms of mutation burden (Supp Figure S1A). Can the authors add some discussion as to why this may be the case?

Response: Thanks for your comments. In the revised manuscript, we systematically evaluated the heterogeneity of capture platforms, and decided to remove the TMB value in the part of results.

1, About PMID30012096

The dataset of PMID30012096 used Agilent SureSelect All Exon V5 capture platform and only included 9 patients (the details were listed below). We could see that 3 of the 9 patients (ZH12, ZH22, ZH24) had quite few SNVs which might be due to the heterogeneity in ESCC genomes.

Patients	Age	Stage	Location	Total_SNV	Exon_SNVs	Nonsynonymous_SNV
PMID30012096;ZH12	70-75	IIB	Middle	9	7	4
PMID30012096;ZH16	55-60	IIB	Middle	69	56	35
PMID30012096;ZH17	55-60	IIB	Middle	88	63	40
PMID30012096;ZH19	50-55	IIB	Upper	92	74	48
PMID30012096;ZH22	55-60	IIB	Middle	3	2	2
PMID30012096;ZH23	75-80	IIB	Upper	129	97	68
PMID30012096;ZH24	40-45	IIB	Middle	8	5	2
PMID30012096;ZH25	55-60	IIB	Middle	350	268	182
PMID30012096;ZH26	45-50	IIB	Middle	60	40	28

On the other side, the chance of stochastic sampling error was higher in small sample size (n=9). The case might be a sampling error.

2, The heterogeneity of capture platforms

In the revised manuscript, we have systematically assessed the influence of capture platforms in methods and sFigure2.

3, We did not provide the TMB value in the revised manuscript

Due to the heterogeneity in the sequence methods of our included studies, we did not provide the estimation of tumor mutational burden (TMB), which was greatly influenced by its denominator of total capture length and would be misleading in direct comparisons between different platforms.

If we use a common subset region of the six capture platforms, the number of non-silent SNVs was less influenced but the total capture length was smaller than common capture platform, thus result in high estimation of TMB value compared with other literatures.

We provided an example of TMB calculation within a specified capture platform in our github repository and encouraged readers to estimate the TMB value based on their own target region.

7. The pathway work in Figure 3 is interesting but could be improved. The pathway analysis on Line 174 says “Although most genes presented low mutational frequency in ESCC, their related functions were enriched in several major oncogenic pathways”. The current analysis does not support this statement as it does not show that the pathways were enriched in this data, only that they are frequently mutated. This could be caused by the pathways having a large number of genes, or the genes in these pathways being large and more likely mutated. Additionally “Around 14.7% patients carried at least targetable mutations, such as BRCA1/2 (5%) and EGFR (2%)” (sFigure3A)”. How were mutations classified as “targetable”?

Response: thanks for your professional comment, we have revised the manuscript.

The “targetable mutations” here should be “druggable mutations”. We collected 14 genes whose mutations had recommended target drugs (sTable8, the following table). These genes were also commonly included in current oncological testing panels. We conceptually defined the non-silent mutations among the 14 genes as “druggable mutations”.

Mutated Gene	FDA Approved Drug
BRCA1	olaparib
BRCA2	olaparib
EGFR	mobocertinib
VEGFA	bevacizumab
ROS1	crizotinib;entrectinib
ALK	crizotinib;ceritinib
BRAF	vemurafenib; dabrafenib
KRAS	sotorasib
NTRK1	larotrectinib
MET	capmatinib
CD274	atezolizumab; avelumab
RET	selpercatinib
ERBB2	afatinib
FGFR1	erdafitinib

8. Figure 3D shows that drinkers and smokers have more EP300 changes and this more hotspot changes. Could this just be due to a higher mutation load in the patients who are drinkers and smokers? Line 213 says: "and associated with worse prognosis compared with mutations in other EP300 regions or wide types (Figure3E)", note the EP300 mutations were only linked to prognosis in late-stage disease.

Response: Thanks for your careful comments. In our integrated dataset, the drinkers and smokers present higher mutational rate in EP300, while the patients of drinkers or smokers did not have higher mutational load (the following figure). The median values of nonsynonymous mutations were 79/85 in somker/non-smoker ($p=0.031$), and 81/84 in drinker/non-drinker($p=0.45$).

We have revised the sentence.

9. Some of the results have changed in the revised version. Previously the authors reported on 1616 samples, and now there are 1930 samples, also "The median number of nonsynonymous mutations in the ESCC-META was 74 (Figure 1A), which corresponded to the estimated median tumor mutation burden (TMB) of 2.07 mutation per Mb". The revised version says: "The median number of nonsynonymous mutations in the ESCC-META was 81 (52 of 25th percentiles and 117 of 75th percentiles), which corresponded to the estimated median tumor mutation burden (TMB) of 2.26 mutation per Mb". What is the reason for these changes? Is it because of the revised number of samples and studies and used?

Response: Thanks for your careful comments. The major reason of this change was the increased size of total samples. However, we had removed the calculation of TMB in the revised manuscript.

The changes between the first version and the updated version of ESCC-META dataset were explained as follows.

1, The currently dataset included 314 newly added patients compared to previous dataset, which changed the overall proportions of tumor stages.

As shown in the following table, the proportion of late-stage tumors (stage III or IV): was 36.7% in the previous version and 38.5% in the current version.

	Previous (1616)	Current (1930)
Stage I	155(9.6%)	193(10%)
Stage II	520(32.2%)	598(31.0%)
Stage III	553(34.2%)	681(35.3%)
Stage IV	40(2.4%)	63(3.2%)
N.A.	348(21.5%)	395(20.5%)

2, We had updated some source data in first ESCC-META dataset (listed in the following table).

Previous	Samples	Current
PMID24686850	19	SRP033394
PMID28365443	18(WES) 7(WGS)	SRP072858_WES SRP072858_WGS
PMID24670651	71(WES) 17(WGS)	SRP034680_WES SRP034680_WGS

These genomes had been included in the previous dataset as their reported article (using their published SNVs records), while we adopted their raw sequence data and performed de novo analysis according to our unified pipelines in the current dataset. This update caused few changes in these samples.

3, We had updated our analysis pipelines in the previous revision.

The reported SNVs from different source often presented different recording format. For example, there could be four types of representation of an identical SNV of two base deletions (the following figure).

Normal	C	T	C	G	T	C
Tumor	C	T	--		T	C
chr2	3	4	5 6		7	8

	POS	REF	ALT
Type 1	chr2:4	TCG	T
Type 2	chr2:5	CG	-
Type 3	chr2:3	TCG	T
Type 4	chr2:4	CG	-

We performed more rigorous genomic check before genomic liftover and annotations in processing heterogeneous SNVs records. This improvement restored some SNVs that had failed in correct annotation because of mismatching (about 0.1% in total SNVs).

The ESCC-META dataset was not changed in this round of revision.

10. Another change is the opening sentence of the results which says “We established a set of pipelines for data selection and process to build integrated 55 genomic cohort (see Methods for details),” If establishment of these pipelines is a key result then the pipelines and

code should be made available. A key feature of this manuscript is the collation of the somatic mutations for the 1930 cases, it would be very helpful if this made available to the readers.

Response: Thanks for your comments. Descriptive explanations of the pipelines were presented in Methods part. The functions and source code of the pipelines would be open in our GitHub repo after all the revision works.

11. The description of the 8 gene signature is important for the novelty of the paper and the inclusion of the validation data for the 8 gene signature has improved the paper. However, due to the importance of the signature in the paper, it would be really good if the authors could address my original comment and add to the discussion how this would be implemented clinically and whether it is better than any clinical or other markers of prognosis that may be used.

Response: Thanks for your advice. We had added explanation of the 8 gene model in the results and discussion part of the revised manuscript.

To recognize and adjust potential confounding of clinical variables, we evaluated the mutational score in multivariable Cox regression. The results proved mutational score as an independent prognostic predictor, and compared with 0 value of the score, one mutation and two or more mutations implied 1.53 [1.29-1.8] and 2.17 [1.63-2.9] of multivariable adjusted HR [95% CI] (the following figure, sFigure6 in revised manuscript).

We added the discussion in the revised manuscript

We proposed the concept of mutational score that combined multiple significant genes as a test panel to increase the positive proportion in test. This model was based on large population and specifically designed for ESCC. Compared with previous reported prediction models, which were often conceptual or theoretical, our work had advantage in robustness and practicality. Owing to the limited involved genes and simplicity of its algorithm, the capture probes for the eight gene mutational score was also applicable for low abundance DNA libraries, such as circulating tumor DNA (ctDNA) sequence in ESCC. Since the mutational score could distinguish the patients with worse prognosis, its dynamic monitoring in ctDNA would be helpful in individualized

treatment.

Other comments

1. Line 268 says "The mutational rate of CSMD3, PCLO, NFE2L2 and FLG were significantly higher in ESCC of upper thoracic part, while the mutation of NOTCH1, MUC4 were more common in lower thoracic part (sTable7)", please add a pvalue between these group to show significance

2. My previous comment asked how the groups of "low" were and "high" TMB were defined. Although this was answered in the rebuttal, please add this to the relevant figure legends, e.g. in Figure S1D.

3. Please review the manuscript for readability and grammar throughout the manuscript. For example:

a) Line 59 says "were re-analysis from raw reads data", should read: were re-analysed from raw reads data

b) Line 101 says "Forest plot of the of the", remove: of the, which is listed twice

c) Line 214 says "wide types", should be: wild types

d) Line 285 says "sFigur4C" and line 286 says "Figur4E", replace with: Figure

e) Line 338 says "ESCC clinical trial (xxxxxxx)" – please correct this text to specify what is meant by xxxxxxxx

f) Line 436 says "phase III clinical trial of xxxxxxxx" – please correct this text to specify what is meant by xxxxxxxx

g) Line 507 says "The MutSig2 were used", should read: The MutSig2 approach was used

h) Line 508 says "were exclude", should read: were excluded

i) Line 509 says "non-silence mutations", please change to: non-silent

j) Line 516 says "to decomposes major mutational signature", should read: to decompose major mutational signatures

k) Line 517 says "The optimal number of separations was selected both considering the most decline in cophenetic correlations and the afford size of residual sum of squares" – the meaning of this sentence is not clear, please revise

l) Line 523 says "The COSMIC Mutational Signatures database (v2) were", should read: The COSMIC Mutational Signatures database (v2) was

m) Line 557 says "The meta-analysis of single proportions in random effect model was 558 also employed to systematically assess the pooled mutational rate of single gene", – the meaning of this sentence is not clear, please revise

n) Please review and edit the rest of the manuscript, in particular the methods section.

Response : Thanks for your comments. We had corrected all the above errors and made other improvements on writing with the help of a language specialist.

For 1, we had presented pvalues in sTable12 and labeled stars for significance in Figure6A and Figure6C. We also revised the testing methods (using Fisher's exact tests instead of regression methods in comparisons of location) and results here in consideration of rigorosity and uniformity.

Hugo Symbol	Age						Location									pvalue	
	yong			old			upper			middle			lower				
	total	mutation	freq	total	mutation	freq	total	mutation	freq	total	mutation	freq	total	mutation	freq		
TP53	210	166	79.05%	242	196	80.99%	0.6377	94	82	87.23%	683	530	77.60%	442	368	83.26%	0.0149
TTN	210	81	38.57%	242	76	31.40%	0.1149	94	33	35.11%	683	229	33.53%	442	162	36.65%	0.5560
MUC16	210	29	13.81%	242	43	17.77%	0.3027	94	11	11.70%	683	106	15.61%	442	69	15.61%	0.6197
NOTCH1	210	21	10.00%	242	53	21.90%	0.0008	94	14	14.89%	683	101	14.79%	442	82	18.55%	0.2370
CSMD3	210	30	14.29%	242	38	15.70%	0.8944	94	19	20.21%	683	91	13.32%	442	63	14.25%	0.1995
KMT2D	210	19	9.05%	242	33	13.64%	0.1410	94	13	13.83%	683	120	17.57%	442	56	12.67%	0.0787
FAT1	210	19	9.05%	242	31	12.81%	0.2305	94	11	11.70%	683	83	12.15%	442	60	13.57%	0.7752
LRP1B	210	17	8.10%	242	25	10.33%	0.5163	94	11	11.70%	683	65	9.52%	442	43	9.73%	0.7629
PCLO	210	22	10.48%	242	22	9.09%	0.6365	94	14	14.89%	683	66	9.66%	442	42	9.50%	0.2641
SYNE1	210	19	9.05%	242	12	4.96%	0.0954	94	10	10.64%	683	57	8.35%	442	37	8.37%	0.7167
DNAH5	210	19	9.05%	242	20	8.26%	0.8669	94	14	14.89%	683	56	8.20%	442	37	8.37%	0.1069
USH2A	210	16	7.62%	242	23	9.50%	0.5061	94	11	11.70%	683	63	9.22%	442	28	6.33%	0.0996
XIRP2	210	7	3.33%	242	24	9.92%	0.0081	94	8	8.51%	683	58	8.49%	442	38	8.60%	1.0000
ZNF750	210	17	8.10%	242	18	7.44%	0.8607	94	10	10.64%	683	49	7.17%	442	45	10.18%	0.1429
FSIP2	210	21	10.00%	242	25	10.33%	1.0000	94	13	13.83%	683	62	9.08%	442	35	7.92%	0.1859
CDKN2A	210	15	7.14%	242	26	10.74%	0.1936	94	12	12.77%	683	56	8.20%	442	41	9.28%	0.3148
RYR2	210	17	8.10%	242	18	7.44%	0.8607	94	8	8.51%	683	58	8.49%	442	36	8.14%	0.9735
NFE2L2	210	19	9.05%	242	20	8.26%	0.8669	94	15	15.96%	683	52	7.61%	442	35	7.92%	0.0318
CSMD1	210	12	5.71%	242	17	7.02%	0.7010	94	4	4.26%	683	58	8.49%	442	21	4.75%	0.0341
PKHD1L1	210	21	10.00%	242	11	4.55%	0.0276	94	7	7.45%	683	49	7.17%	442	37	8.37%	0.7783
OBSCN	210	17	8.10%	242	21	8.68%	0.8664	94	7	7.45%	683	46	6.73%	442	21	4.75%	0.3097
ZFX4	210	14	6.67%	242	16	6.61%	1.0000	94	10	10.64%	683	42	6.15%	442	25	5.66%	0.1936
EP300	210	15	7.14%	242	15	6.20%	0.7087	94	4	4.26%	683	60	8.78%	442	35	7.92%	0.3411
PIK3CA	210	10	4.76%	242	14	5.79%	0.6787	94	7	7.45%	683	41	6.00%	442	36	8.14%	0.3650

The revised sentence says:

The mutational rate of NFE2L2, TEP1, DMXL1 and NOS1 were higher in upper thoracic part, while the mutations of MUC16, NOTCH1 were more common in lower thoracic part (Figure6C, sTable12).

For 2, we used ordinal groups and labelled them on top in the revised manuscript (Figure1E).

We could see that no significant difference between groups. In fact, we could not obtain different survival groups in any cut off value:

For k) point, the sentence had revised to:

The optimal number of separations (k) was selected both considering the cophenetic correlations and the residual sum of squares (RSS). We chose 11 as the best number of separation because the cophenetic correlations presented maximum decrease between $k=11$ and $k=12$, while the declines of RSS were obviously slower after $k=11$ (Figure2C).

For m) point, the sentence had revised to:

The random effect model in meta-analysis was employed to estimate the inverse variance weighted pooled mutational rates

For the e) and f) points (clinical trial of xxxxxxxx), we had intentionally hidden the registration number of the clinical trial in the manuscript for review because of the requirement of double-blind review. The other researcher related information including ethical statement, data availability and code availability was also hidden. We had explained

these details in the cover letter and a special email to editors. The actual information would be provided in final accepted version.

Reviewers' Comments:

Reviewer #4:

Remarks to the Author:

For full disclosure at the outset, this review is somewhat unusual because the editors have asked me to fill in for the original reviewer 2, who was unable to continue reviewing this manuscript for some reason. Mostly, I will be verifying the authors' responses to the previous comments by the original reviewer. I think the authors have answered the previous comments sufficiently, for the most part. Some specific points I would like to mention:

1. The previous reviewer asked why there were so many statistically significant genes from MutSigCV analysis. I did not see the previous version of this manuscript, but I agree with this reviewer that >800 significant genes does seem surprisingly high. Taking a closer at the MutSigCV overview page at <https://www.genepattern.org/modules/docs/MutSigCV>, I noticed this:

"It has been observed that MutSigCV may not produce useful results on cancers with low mutation rates (such as pediatric cancers) due to certain internal assumptions made in the code. While a future version of MutSigCV may add the ability to change these assumptions before running the analysis, at present the GenePattern module is limited to using these defaults. It is possible to work around these assumptions, though you will need to obtain and modify the MatLab code and run it outside of GenePattern. Please contact the MutSigCV authors for more details."

If MutSigCV might be producing unreliable results because ESCC samples have low mutation loads, then that should be noted in the paper so readers know to interpret those results with some caution.

2. In their response to previous reviewer's comments about small indels, the authors wrote: "We had noticed significant heterogeneity in small insertions and deletions (ID), copy number variants (CNV) and structures variants (SV) in our analysis. Due to the lack of effective approaches to suppress these batch effects, we excluded this analysis from our works."

There's value to reporting this as a supplementary figure so that other workers in the field are more aware of these batch effects and someone might figure out what the root cause is.

3. Running HRDetect is a fairly involved process. The authors would need to analyze WGS data de novo from 600+ samples to derive files for HRDetect, so it's a pretty big ask that the previous reviewer requested. HRDetect results would be nice to have, but I don't think it's a critical part of this specific paper.

4. The previous reviewer wrote:

"7. The pathway work in Figure 3 is interesting but could be improved. The pathway analysis on Line 174 says "Although most genes presented low mutational frequency in ESCC, their related functions were enriched in several major oncogenic pathways". The current analysis does not support this statement as it does not show that the pathways were enriched in this data, only that they are frequently mutated. This could be caused by the pathways having a large number of genes, or the genes in these pathways being large and more likely mutated."

I'm a bit confused. Was this addressed? Is this now Figure 6D? The figure legend needs more detail. What is the x-axis showing? What is considered statistically significant? Many of upper part prone groups have $p.adjust > 0.05$, are those considered significant also? This figure is somewhat misleading because it's the same color scale for both panels but the range of the upper one is roughly an order of magnitude bigger than in the lower panel. Should put this on one unified color scale, I think.

I also have some comments of my own, which should not be significant hurdles to address:

5. Throughout the manuscript, the terms "mutational frequency" and "mutational rate" appear to be used interchangeably. Frequency is simply number of mutants divided by a population count, but rate is probability of mutation per unit time. Frequency is straightforward to measure, but rate estimation can be considerably more debateable. As far as I can tell, the data in this manuscript are frequencies and this should be corrected.

6. There are references to both cosine difference and cosine similarity. To avoid possible confusion to readers, just pick one measure and stick with it.

7. The mutational signature analysis should be upgraded to use COSMIC v3 reference signatures. Your sig10 is a strong match for SBS33 (cosine similarity = 0.96). And sig11 is a closer match to SBS44 (cosine similarity = 0.88), which is a DNA mismatch repair signature. Neither SBS33 nor SBS44 are present in COSMIC v2 reference signatures.

8. sig9 is a very close match to SBS22 (cosine similarity = 0.98) from aristolochic acid, which makes complete sense because of cancer site and patient population. Certainly should be mentioned in the manuscript.

Reviewer #4 (Remarks to the Author):

For full disclosure at the outset, this review is somewhat unusual because the editors have asked me to fill in for the original reviewer 2, who was unable to continue reviewing this manuscript for some reason. Mostly, I will be verifying the authors' responses to the previous comments by the original reviewer. I think the authors have answered the previous comments sufficiently, for the most part. Some specific points I would like to mention:

1. The previous reviewer asked why there were so many statistically significant genes from MutSigCV analysis. I did not see the previous version of this manuscript, but I agree with this reviewer that >800 significant genes does seem surprisingly high. Taking a closer at the MutSigCV overview page at <https://www.genepattern.org/modules/docs/MutSigCV>, I noticed this:

"It has been observed that MutSigCV may not produce useful results on cancers with low mutation rates (such as pediatric cancers) due to certain internal assumptions made in the code. While a future version of MutSigCV may add the ability to change these assumptions before running the analysis, at present the GenePattern module is limited to using these defaults. It is possible to work around these assumptions, though you will need to obtain and modify the MatLab code and run it outside of GenePattern. Please contact the MutSigCV authors for more details."

If MutSigCV might be producing unreliable results because ESCC samples have low mutation loads, then that should be noted in the paper so readers know to interpret those results with some caution.

Response: Thanks for your professional advice, we have reviewed the code and our results. We totally agree with your comments and added this note in the Methods part. The limitation of single method could be overcome by combined approaches to jointly evaluate the significance of mutated genes, which is strategy we used in the study.

2. In their response to previous reviewer's comments about small indels, the authors wrote: "We had noticed significant heterogeneity in small insertions and deletions (ID), copy number variants (CNV) and structures variants (SV) in our analysis. Due to the lack of effective approaches to suppress these batch effects, we excluded this analysis from our works."

There's value to reporting this as a supplementary figure so that other workers in the field are more aware of these batch effects and someone might figure out what the root cause is.

Response: Thanks for your advice. We have added the results of small insertions and deletions (ID83 profile) of WGS genomes in sFigure3 a, b, and the source data are provided in Source Data file.

During the period of this revision, we noticed that the COSMIC database published the signatures of copy number variations (CN) in June 2022, which uses the 48-channel copy number classification scheme and would be more stable across experimental platforms.

<https://cancer.sanger.ac.uk/signatures/cn/>

<https://www.nature.com/articles/s41586-022-04738-6>

We are trying to update the CNV analysis with the latest published methods, and the relevant results would be upload to the public data repository of ESCC-META. But for cautious consideration, we do not hope to add these exploratory results to this manuscript.

3. Running HRDetect is a fairly involved process. The authors would need to analyze WGS data de novo from 600+ samples to derive files for HRDetect, so it's a pretty big ask that the previous reviewer requested. HRDetect results would be nice to have, but I don't think it's a critical part of this specific paper.

Response: Thanks for your understanding. We are trying to do the genomic stability analysis including HRDetect, and would update the results in the online data repository of ESCC-META.

4. The previous reviewer wrote:

"7. The pathway work in Figure 3 is interesting but could be improved. The pathway analysis on Line 174 says "Although most genes presented low mutational frequency in ESCC, their related functions were enriched in several major oncogenic pathways". The current analysis does not support this statement as it does not show that the pathways were enriched in this data, only that they are frequently mutated. This could be caused by the pathways having a large number of genes, or the genes in these pathways being large and more likely mutated."

I'm a bit confused. Was this addressed? Is this now Figure 6D? The figure legend needs more detail. What is the x-axis showing? What is considered statistically significant? Many of upper part prone groups have p.adjust > 0.05, are those considered significant also? This figure is somewhat misleading because it's the same color scale for both panels but the range of the upper one is roughly an order of magnitude bigger than in the lower panel. Should put this on one unified color scale, I think.

Response:

The comment 7 of previous reviewer referred to the Figure 3 and Figure 4 in the revised version.

The old sentence of

“Although most genes presented low mutational frequency in ESCC, their related functions were enriched in several major oncogenic pathways”

is not accurate, because we just summarized the mutated genes by related pathways and did not perform enrichment analysis.

In the previous revision, we changed it as

“We summarized the mutated genes by their related oncogenic pathways (Figure 3a) and found that that 38.1% ESCC patients had at least one mutation in Hippo pathway (including FAT1, FAT2, FAT3), 38.6% in histone modification, 33.8% in NOCTH pathway (KMT2D, KMT2C, EP300, CREBBP), 19.8% in RTK-RAS pathway (ERBB4 and ROS1), 17.6% in cell cycle pathway (CDKN2A, RB1), 15.3% in PI3K pathway (PIK3CA), and 12.6% in Nrf2 pathway (NFE2L2, KEAP1).”

We agree that the Figure 6D is somewhat misleading, and we have modified it in the revised manuscript.

In the revised Figure 6d, the top 15 enriched pathways from GO analysis of upper part prone genes (upper part) or lower part prone genes (lower part) were shown. The horizontal axis indicates the value of $-\log_{10}(p.adjust)$ in GO analysis, and the labeled * represents for $p (adjusted) < 0.05$, ** for $p (adjusted) < 0.01$.

I also have some comments of my own, which should not be significant hurdles to address:

5. Throughout the manuscript, the terms "mutational frequency" and "mutational rate" appear to be used interchangeably. Frequency is simply number of mutants divided by a population count, but rate is probability of mutation per unit time. Frequency is straightforward to measure, but rate estimation can be considerably more debateable. As far as I can tell, the data in this manuscript are frequencies and this should be corrected.

Response: Thanks for your professional advice. We unified use "*mutational frequency*" in the revised manuscript.

6. There are references to both cosine difference and cosine similarity. To avoid possible confusion to readers, just pick one measure and stick with it.

Response: Thanks for your advice. We unified use cosine similarity in the revised manuscript.

7. The mutational signature analysis should be upgraded to use COSMIC v3 reference signatures. Your sig10 is a strong match for SBS33 (cosine similarity = 0.96). And sig11 is a closer match to SBS44 (cosine similarity = 0.88), which is a DNA mismatch repair signature. Neither SBS33 nor SBS44 are present in COSMIC v2 reference signatures.

Response: Thanks for your advice. We have upgraded to COSMIC3 in the revised manuscript.

8. sig9 is a very close match to SBS22 (cosine similarity = 0.98) from aristolochic acid, which makes complete sense because of cancer site and patient population. Certainly should be mentioned in the manuscript.

Response: Thanks for your professional advice. We added this point in the result.

“There were 1.2% patients (n=24) presented prevalent mutational pattern of sig9 or SBS22 (similarity = 0.98), which was associated to aristolochic acid exposure, and thus suggested the specific carcinogenesis in this subgroup patients^{27,28}.”

27	Lim, A. H. et al. Rare Occurrence of Aristolochic Acid Mutational Signatures in Oro-Gastrointestinal Tract Cancers. Cancers (Basel) 14 , doi:10.3390/cancers14030576 (2022). ⁴²
28	Poon, S. L. et al. Genome-wide mutational signatures of aristolochic acid and its application as a screening tool. Sci Transl Med 5 , 197ra101, doi:10.1126/scitranslmed.3006086 (2013). ⁴²